# Prior-Guided Flow Matching for Target-Aware Molecule Design with Learnable Atom Number

**Jingyuan Zhou**[1], **Hao Qian**[1], **Shikui Tu** [1*], **Lei Xu**[1,2*]

[1]School of Computer Science, Shanghai Jiao Tong University
[2]Guangdong Laboratory of Artificial Intelligence and Digital Economy (SZ)
`{zjoyuan0930, qhonearth, tushikui, leixu}@sjtu.edu.cn`

## Abstract

Structure-based drug design (SBDD), aiming to generate 3D molecules with high binding affinity toward target proteins, is a vital approach in novel drug discovery. Although recent generative models have shown great potential, they suffer from unstable probability dynamics and mismatch between generated molecule size and the protein pockets geometry, resulting in inconsistent quality and off-target effects. We propose PAFlow, a novel target-aware molecular generation model featuring prior interaction guidance and a learnable atom number predictor. PAFlow adopts the efficient flow matching framework to model the generation process and constructs a new form of conditional flow matching for discrete atom types. A protein–ligand interaction predictor is incorporated to guide the vector field toward higher-affinity regions during generation, while an atom number predictor based on protein pocket information is designed to better align generated molecule size with target geometry. Extensive experiments on the CrossDocked2020 benchmark show that PAFlow achieves a new state-of-the-art in binding affinity (up to -8.31 Avg. Vina Score), simultaneously maintains favorable molecular properties. The code of the paper is provided at https://github.com/CMACH508/PAFlow.

## 1 Introduction

As a subfield of deep learning in drug discovery, structure-based drug design (SBDD) is considered as a rational and challenging approach for developing novel drugs [1, 2]. It aims to generate drug-like molecules with stable 3D structures and high binding affinities conditioned on the target proteins, which can be formulated as a conditional generation problem. In the past few years, an increasing number of generative models have been applied to SBDD task. One category involves autoregressive models that generate 3D molecules atom by atom [3–5] or fragment by fragment [6, 7]. Another category includes diffusion-based methods [8–11], which gradually denoise standard Gaussian noise to predict full-atom distributions by leveraging both local and global information. Other models, such as Rectified Flow [12] and Bayesian Flow Networks [13], have also been employed in SBDD [14, 15]. Although these methods have demonstrated the potential to generate target-aware molecules, they still face several limitations: (1) autoregressive sampling suffers from unnatural generation orders, leading to unrealistic fragments and error accumulation; (2) the denoising trajectory in diffusion models is highly stochastic, resulting in unstable molecular quality; (3) to the best of our knowledge, all current non-autoregressive methods determine atom numbers in generated molecules by sampling from a predefined distribution, which relies on prior knowledge from reference ligands and often causes mismatches between ligand size and binding site geometry.

---

[*]Correspondence authors are Shikui Tu and Lei Xu.

39th Conference on Neural Information Processing Systems (NeurIPS 2025).

The recently proposed Flow Matching (FM) framework [16] has shown promising initial results and demonstrates potential to serve as an effective solution for SBDD problems. Specifically, FM is a simulation-free approach for training Continuous Normalizing Flows (CNFs) that exhibits notable generative capabilities. It is compatible with the Gaussian probability paths employed in diffusion models for transitioning between noise and data samples, while also supporting non-Gaussian paths as conditional probability trajectories. During the sampling phase, the adoption of ordinary differential equation (ODE) solvers enables fast and stable generation.

Inspired by the recent advancement of FM, we propose PAFlow, a 3D all-atom **Flow** matching model with **P**rior interaction guidance and a learnable **A**tom number predictor, which is designed to address the above drawbacks of existing methods. To overcome limitations (1) and (2), PAFlow adopts the FM as the framework for molecule generation. Specifically, we employ the established Variance Preserving (VP) probability path [16] for generating continuous atomic coordinates while deriving a new form of Conditional Flow Matching (CFM) for discrete atom type. To further enhance the binding affinity between molecules and target proteins, a protein–ligand interaction predictor is incorporated during generation, which guides the vector field using prior binding knowledge towards poses with tighter interactions. Regarding limitation (3), we develop an atom number predictor that only utilizes protein pocket information rather than reference ligand priors to estimate the appropriate number of atoms. Extensive empirical evaluation on the CrossDocked2020 dataset [17] demonstrates that PAFlow generates molecules with not only significantly improved binding affinity compared to all baseline methods but also maintaining desirable molecular properties.

Our main contributions can be summarized as follows:

- We propose an SBDD generative model based on the FM framework, where atomic coordinates and atom types are modeled using the existing VP path and a newly developed CFM, respectively.

- A protein-ligand interaction predictor is integrated into the generation process to introduce prior binding knowledge, guiding the vector field toward directions that correspond to higher binding affinity.

- To address the mismatch between molecule size and binding site geometry when sampling from the predefined distributions, we introduce an atom number predictor that estimates the appropriate number of atoms only using the binding site information.

- Extensive experiments on the CrossDocked2020 benchmark demonstrate that PAFlow can generate molecules with **-8.31 Avg. Vina Score**, significantly outperforming other strong baselines by a large margin (-1.24) and establishing a new state-of-the-art, while maintaining favorable molecular properties.

## 2    Related Works

**Structure-Based Drug Design**    Structure-based drug design is a fundamental task in drug discovery, which aims to generate molecules that specifically bind with high affinity to a given protein pocket [18]. Early efforts such as [19, 20] generate 1D SMILES strings based on protein context, while [21] fit the distribution of target sequence embeddings in latent space to enable target-aware molecular graph generation. Motivated by the advances in 3D and geometric modeling, numerous works have attempted to address the problem directly in the 3D space. [22] voxelizes molecules in atomic density grids to generate 3D molecules in a conditional VAE framework. [3–5] propose to iteratively generate atoms and bonds through autoregressive sampling within target binding site. [6, 7] generate ligand molecules motif by motif utilizing chemical priors of molecular fragments. However, the unnatural generation order during autoregressive sampling leads to unrealistic fragments and severe error accumulation. In recent works, diffusion models have been extensively applied to ligand molecule generation, which denoises atom types and coordinates sampled from prior distributions using SE(3)-equivariant networks [8–11, 23, 24]. [15] also explore the application of Bayesian Flow Networks (BFN) for molecular generation. However, these non-autoregressive methods typically acquire the number of atoms in the generated molecules by sampling from a predefined distribution, which depends on reference ligand information and often results in mismatches between the molecular size and the protein pocket. In this work, we aim to address this issue by introducing a learnable atom number predictor.

**Flow Matching** Flow Matching has recently attracted considerable attention and has demonstrated its potential in various domains such as image generation [25, 26] and biomolecule design [27, 28]. As a simulation-free approach for training continuous normalizing flows (CNFs), it is compatible with diverse probability paths and achieves better sampling efficiency through ODE solvers. For instance, [16] apply Flow Matching to the Variance Preserving diffusion path and formulate the VP Conditional Flow Matching (CFM), which addresses the unstable probability dynamics in conventional diffusion models. Other works [12, 26, 29] have explored transporting the prior distribution to the data distribution along straight line paths as much as possible. Although such paths offer shorter distances and lower computational cost, they lack the capacity to effectively model complex tasks like SBDD. For instance, [14] adopt this strategy for SBDD but achieve suboptimal performance. Therefore, we adopt the more robust VP path to model continuous atomic coordinates and construct a novel CFM tailored for discrete atomic types.

## 3 Preliminaries

**Problem Definition** The SBDD task can be defined as a conditional generative problem, where the goal is to generate ligand molecules that specifically bind to a given protein target. A target protein $\mathcal{P}$ is represented as a set of atoms $\mathcal{P} = \{(\mathbf{x}_P^{(i)}, \mathbf{a}_P^{(i)})\}_{i=1}^{N_P}$, where $\mathbf{x}_P^{(i)} \in \mathbb{R}^3$ is the 3D coordinates of the $i$-th protein atom, $\mathbf{a}_P^{(i)} \in \mathbb{R}^{N_f}$ is a one-hot vector representing its features (e.g., element type, amino acid type), $N_P$ is the number of atoms contained in the protein and $N_f$ is the feature dimension of a protein atom. Correspondingly, a binding molecule $\mathcal{M} = \{(\mathbf{x}_M^{(i)}, \mathbf{a}_M^{(i)})\}_{i=1}^{N_M}$ comprises $N_M$ atoms with 3D coordinates $\mathbf{x}_M^{(i)} \in \mathbb{R}^3$ and atom types $\mathbf{a}_M^{(i)} \in \mathbb{R}^K$, where $K$ is the molecule atom type dimension. Then molecular representation can be simplified as $\mathbf{m} = [\mathbf{x}_M, \mathbf{a}_M]$, where $\mathbf{x}_M \in \mathbb{R}^{N_M \times 3}$, $\mathbf{a}_M \in \mathbb{R}^{N_M \times K}$ and $[\cdot, \cdot]$ is the concatenation operator. Similarly, the binding pocket is denoted as $\mathbf{p} = [\mathbf{x}_P, \mathbf{a}_P]$, where $\mathbf{x}_P \in \mathbb{R}^{N_P \times 3}$ and $\mathbf{a}_P \in \mathbb{R}^{N_P \times N_f}$.

**Preliminaries on Flow Matching** In this section, we summarize how the general flow matching method is implemented based on [16]. Let $\mathbb{R}^d$ represent the data space with data points $x = (x^1, \ldots, x^d) \in \mathbb{R}^d$. $q$ is the data distribution, $x_1$ denotes a data point from $q$ and $x_0$ represents a sample from the prior distribution $p_0$. The time-dependent probability density path is defined as $p_{t \in [0,1]} : \mathbb{R}^d \to \mathbb{R}_{>0}$ and the corresponding vector field is defined as $u_{t \in [0,1]} : \mathbb{R}^d \to \mathbb{R}^d$. The vector field can construct a unique time-dependent flow $\psi_{t \in [0,1]} : \mathbb{R}^d \to \mathbb{R}^d$ defined by the ODE:

$$\frac{\mathrm{d}}{\mathrm{d}t} \psi_t(x) = u_t(\psi_t(x)), \quad \psi_1(x) = x. \tag{1}$$

FM seeks to regress the target vector field $u_t$ with a neural network $v_\theta(x, t)$:

$$\mathcal{L}_{FM}(\theta) = \mathbb{E}_{t, p_t(x)} \| v_\theta(x, t) - u_t(x) \|^2. \tag{2}$$

However, the lack of knowledge about what an appropriate $p_t$ and $u_t$ are makes Eq. 2 infeasible to apply in practice. [16] proposed an alternative that employs a definable conditional probability path $p_t(x|x_1)$ with a conditional vector field $u_t(x|x_1)$. The Conditional Flow Matching (CFM) objective is formulated as follows:

$$\mathcal{L}_{CFM} = \mathbb{E}_{t, p_1(x_1), p_t(x|x_1)} \| v_\theta(x, t) - u_t(x|x_1) \|^2. \tag{3}$$

The CFM objective is tractable and shares the same gradient as $\mathcal{L}_{FM}$ [16, 30], thus optimizing the CFM objective is equivalent to optimizing the FM objective. As for inference phase, Eq. 1 can be solved using ODE solvers: $x_1 = \mathrm{ODESolve}(x_0, v_\theta, 1, 0)$.

## 4 Method

This section elaborates on the implementation details of PAFlow. First, different probability paths are used to model continuous atomic coordinates and discrete atom types within the FM framework while maintaining SE(3)-equivariance during ODE-based sampling. Based on this backbone, a protein–ligand interaction predictor is employed guide the vector field during generation to further enhance molecular quality. Additionally, to address the geometric incompatibility between generated molecules and target proteins, an atom number predictor that relies solely on target protein infor-

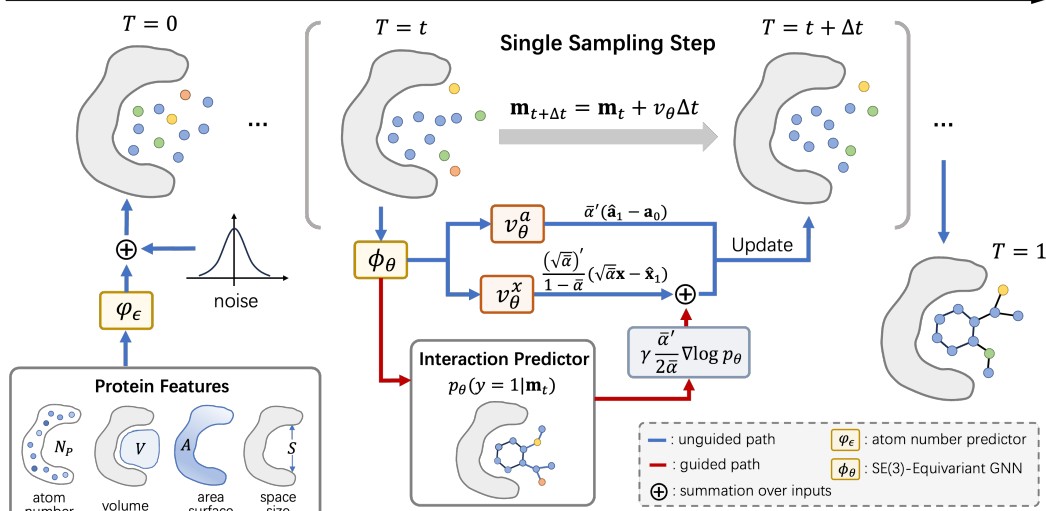

Figure 1: Overview of the PAFlow generation process. The atom predictor first estimates the number of atoms in the generated molecule based on the target protein information, which is then used to initialize the molecule. SE(3)-EGNN is applied to predict vector fields for atomic coordinates and atom types, while a protein-ligand interaction predictor provides prior binding guidance for the coordinate vector field. The final ligand molecule with high binding affinity is obtained through iterative updates. For simplicity, $\bar{\alpha}_{1-t}$ is denoted as $\bar{\alpha}$.

mation—rather than prior knowledge from reference ligands—is proposed. The whole training and sampling procedures are summarized in the Appendix C.

## 4.1 Flow Matching in Molecule Generation

For PAFlow, a data point from the prior distribution $p_0$ is $\mathbf{m}_0$ and the target data distribution $p_1$ is the ligand molecule $\mathbf{m}_1$ that can specifically binds to the target protein. The molecular probability path can be formulated as a product of atom coordinate distribution and atom type distribution following [8]:

$$p_t(\mathbf{m}|\mathbf{m}_1, \mathbf{p}) = p_t(\mathbf{x}|\mathbf{x}_1, \mathbf{p}) \cdot p_t(\mathbf{a}|\mathbf{a}_1, \mathbf{p}). \tag{4}$$

For brevity we use $\mathbf{x}$ in place of $\mathbf{x}_M$. [31] proposes that in the multivariable case the probability path for each variable can be constructed separately, which allows us to model the two variables $\mathbf{x}$ and $\mathbf{a}$—belonging to different data types—using separate probability paths. The continuous atomic coordinates $\mathbf{x}$ are modeled using the Variance Preserving path, with the probability path and target conditional vector field defined as follows [16]:

$$p_t(\mathbf{x}|\mathbf{x}_1, \mathbf{p}) = \mathcal{N}(\mathbf{x}|\sqrt{\bar{\alpha}_{1-t}}\mathbf{x}_1, (1 - \bar{\alpha}_{1-t})\mathbf{I}), \qquad u_t^x(\mathbf{x}|\mathbf{x}_1, \mathbf{p}) = \frac{(\sqrt{\bar{\alpha}_{1-t}})'}{1 - \bar{\alpha}_{1-t}}(\sqrt{\bar{\alpha}_{1-t}}\mathbf{x} - \mathbf{x}_1), \tag{5}$$

where $\mathcal{N}$ is a Gaussian distribution and the fixed variance schedules are defined as $\beta_t$ ($t = 0, \Delta t, ..., 1$), with $\alpha_t = 1 - \beta_t$, $\bar{\alpha}_t = \prod_{s=0}^{t} \alpha_t$, $\bar{\beta}_t = 1 - \bar{\alpha}_t$, following [8]. $(\sqrt{\bar{\alpha}_{1-t}})'$ denotes the derivative with respect to time $t$. For the discrete atomic types $\mathbf{a}$, the probability density path is defined according to [8] as a categorical distribution $\mathcal{C}$:

$$p_t(\mathbf{a}|\mathbf{a}_1, \mathbf{p}) = \mathcal{C}(\mathbf{a}|\mathbf{c}(\mathbf{a}, \mathbf{a}_1)), \qquad where \quad \mathbf{c}(\mathbf{a}, \mathbf{a}_1) = \bar{\alpha}_{1-t}\mathbf{a}_1 + (1 - \bar{\alpha}_{1-t})/K \tag{6}$$

In practice $\mathbf{x}$ and $\mathbf{a}$ follow different schedules, but we retain the same notation for conciseness. Since this probability path has not been constructed as CFM in previous work, we derive its conditional vector field as below:

$$u_t^a(\mathbf{c}(\mathbf{a}, \mathbf{a}_1)|\mathbf{a}_1, \mathbf{p}) = \bar{\alpha}_{1-t}'(\mathbf{a}_1 - \mathbf{a}_0). \tag{7}$$

The complete derivation is given in Appendix A.2. In fact, the probability paths for atom coordinates and types are identical to those applied in [8]; however, the generation strategies adopted by the two methods are fundamentally different. In [8], generation is performed by iteratively denoising

the Gaussian noise. Instead, our work formulates the generation process as the integration of the ODE $\frac{d\mathbf{m}_t}{dt} = v_\theta(\mathbf{m}_t, t)$ from $t = 0$ to $t = 1$ using an Euler solver [32] starting from an initialized ligand molecule $\mathbf{m}_0$, where the vector field is approximated with a neural network parameterized by $\theta$. Specifically, for $\mathbf{x}$ and $\mathbf{a}$ we have:

$$\mathbf{x}_{t+\Delta t} = \mathbf{x}_t + v_\theta^x(\mathbf{m}_t, \mathbf{p}, t)\Delta t, \qquad \mathbf{c}(\mathbf{a}_{t+\Delta t}, \mathbf{a}_1) = \mathbf{c}(\mathbf{a}_t, \mathbf{a}_1) + v_\theta^a(\mathbf{m}_t, \mathbf{p}, t)\Delta t, \qquad (8)$$

where $\mathbf{x}_0$ is initialized with a standard Gaussian distribution inside the protein pocket and $\mathbf{a}_0$ are initialized with a uniform distribution. The generative process is expected to be invariant to translations and rotations of the protein-ligand complex, an essential inductive bias when generating 3D molecules [33–36]. Given the evidence that an invariant distribution composed with an equivariant invertible function will result in an invariant distribution [33], we have the following proposition (proof in Appendix A.1).

**Proposition 1.** *Denoting the SE(3)-transformation as $T_g$ and $(v_\theta^x(\mathbf{m}_t, \mathbf{p}, t), v_\theta^a(\mathbf{m}_t, \mathbf{p}, t)) = v_\theta(\mathbf{m}_t, \mathbf{p}, t)$, if we shift the Center of Mass (CoM) of protein atoms to zero and parameterize $v_\theta(\mathbf{m}_t, \mathbf{p}, t)$ with an SE(3)-equivariant network, then the generation process is invariant w.r.t $T_g$ on the protein-ligand complex.*

There are different ways to parameterize $v_\theta$. In this work the neural network is designed to predict $[\mathbf{x}_1, \mathbf{a}_1]$, which are then processed through Eq. 5 and Eq. 7 to obtain $v_\theta^x$ and $v_\theta^a$ respectively. To guarantee the invariance constraint, the generation process is parameterized using an SE(3)-Equivariant GNN $\phi_\theta$:

$$[\hat{\mathbf{x}}_1, \hat{\mathbf{a}}_1] = \phi_\theta([\mathbf{x}_t, \mathbf{a}_t], t, \mathbf{p}), \qquad (9)$$

where the $l$-th layer works as:

$$\mathbf{h}_i^{l+1} = \mathbf{h}_i^l + \sum_{j \in \mathcal{V}, i \neq j} f_h(d_{ij}^l, \mathbf{h}_i^l, \mathbf{h}_j^l, \mathbf{e}_{ij}; \theta_h) \qquad (10)$$

$$\mathbf{x}_i^{l+1} = \mathbf{x}_i^l + \sum_{j \in \mathcal{V}, i \neq j} (\mathbf{x}_i^l - \mathbf{x}_j^l) f_x(d_{ij}^l, \mathbf{h}_i^{l+1}, \mathbf{h}_j^{l+1}, \mathbf{e}_{ij}; \theta_x) \cdot \mathbf{l}_{mask}. \qquad (11)$$

$l$ denotes the layer index and $\mathcal{V}$ is the set of atoms from both the protein and the ligand molecule. $d_{ij} = ||\mathbf{x}_i - \mathbf{x}_j||$ and $\mathbf{e}_{ij}$ represents the relative distance and option edge features between atom $i$ and atom $j$. $\mathbf{l}_{mask}$ is a mask applied to the ligand atoms to keep the protein atom coordinates fixed. $f_h$ and $f_x$ are graph attention networks. $\hat{\mathbf{a}}_1$ can be obtained by input $\mathbf{h}^L = [\mathbf{h}_1^L, ..., \mathbf{h}_{N_M-1}^L]_t$ into a multi-layer perceptron and a softmax function.

## 4.2 Prior-Guided Generation

Inspired by previous works [23, 10, 24] that utilize prior knowledge to guide the denoising process of diffusion models, we leverage predicted protein-ligand interactions, i.e. binding affinity, as guidance for the vector field, thereby further improving the quality of generated molecules. The ground truth of binding affinity is denoted as $y$. The binding affinity of ligands in the training set is normalized to $[0, 1]$, thus $y = 1$ indicates a strong interaction between the molecule and the target protein. To predict the binding affinity $\hat{y}$ between generated molecule and target protein, the final atom hidden embedding $\mathbf{h}^L$ containing useful global information is used following [8, 23] and Eq. 9 can be rewritten as:

$$[\hat{\mathbf{x}}_1, \hat{\mathbf{a}}_1, \hat{y}] = \phi_\theta([\mathbf{x}_t, \mathbf{a}_t], t, \mathbf{p}), \qquad \hat{y} = \frac{1}{N_M} \sum_{i=0}^{N_M-1} \sigma(\text{MLP}(\mathbf{h}^L)), \qquad (12)$$

where $\sigma$ is a sigmoid function. According to *Lemma 1* from [37], we derive the predictor guidance generation process as follows:

$$\mathbf{x}_{t+\Delta t} = \mathbf{x}_t + (v_\theta^x(\mathbf{m}_t, \mathbf{p}, t) + \gamma \frac{\bar{\alpha}_{1-t}'}{2\bar{\alpha}_{1-t}} \nabla \log p_\theta(y = 1|\mathbf{m}_t))\Delta t, \qquad (13)$$

$$\mathbf{c}(\mathbf{a}_{t+\Delta t}, \mathbf{a}_1) = \mathbf{c}(\mathbf{a}_t, \mathbf{a}_1) + v_\theta^a(\mathbf{m}_t, \mathbf{p}, t)\Delta t. \qquad (14)$$

Detailed derivation can be found in Appendix A.3. A scaling factor $\gamma$ is added to control the gradient strength and the likelihood function can be defined as:

$$p_\theta(y = 1|\mathbf{m}_t) \propto \exp(-\ell(\hat{y}, y = 1)), \qquad (15)$$

where $\ell : \mathbb{R} \times \mathbb{R} \to \mathbb{R}$ is the mean square error (MSE) loss function to evaluate the deviation between the predicted score $\hat{y}$ and $y = 1$.

Explicitly guiding discrete atom types through gradients in molecular generation is challenging. However, in this work, at sampling time step $t$, the sampled atom type $\mathbf{a}_{t+\Delta t}$ is obtained from $\mathbf{x}_t$ and $a_t$ via the SE(3)-Equivariant GNN. As a result, the optimized $\mathbf{x}_t$ naturally influences $\mathbf{a}_{t+\Delta t}$, thereby providing the implicit guidance for atom types. This guidance strategy is consistent with the approach employed in [24].

## 4.3 Learnable Atom Number Predictor

To generate molecules that better match the size of the protein pocket and avoid relying on prior knowledge from reference ligands, an atom number predictor $\varphi_\epsilon$ with neural network is trained using only target protein information. [38] suggests that there is a certain correlation between the volume of the binding pocket $V$ and the ligand size. To provide the network with comprehensive information about the binding pocket and improve prediction accuracy, the surface area of the binding site $A$, the number of atoms within the pocket $N_P$, and the space size $S$—which is used by existing models to determine atom numbers—are also utilized as inputs. Besides, a dataset consisting of binding site information–ligand atom number pairs is constructed for training. To enhance training stability and accelerate convergence, normalized ligand atom numbers are used as training labels, then the network outputs must be denormalized to obtain the final predicted number of atoms:

$$\hat{N}_M = \hat{n}_M(N_M^{max} - N_M^{min}) + N_M^{min}, \qquad \hat{n}_M = \varphi_\epsilon([N_P, V, A, S]) + \tau \qquad (16)$$

[39] propose that introducing noise in generative models can enhance both the diversity and quality of discrete outputs. We observe the improved molecular generation performance when a small Gaussian noise term $\tau \sim N(0, \delta^2)$ is added to the predictor's output, which resembles the reparameterization trick used in variational autoencoders. Neural networks inherently involve uncertainty in their predictions, and the injection of Gaussian noise may reflect this uncertainty. As a form of regularization, this strategy helps mitigate overconfidence in potentially inaccurate point estimates and encourages exploration of a broader solution space. We further prove the effectiveness of noise injection in a mathematical form in Appendix A.4.

## 5 Experiments

### 5.1 Experimental Setup

**Dataset**    We train and evaluate PAFlow on the CrossDocked2020 dataset [17]. Following the same strategy as in [3, 8], binding poses with RMSD greater than 1 Å and protein pairs with sequence identity over 30% are excluded. Similar to [23], binding affinities in the CrossDocked2020 dataset are normalized to the range $[0, 1]$ for training the protein–ligand interaction predictor, where higher values indicate better binding affinity. We then randomly sample 100,000 complexes for training and select 100 complexes with distinct target proteins for testing. We further process the data and construct a dataset in the form of $\mathcal{D} = \{(N_M, N_P, V, A, S)\}$, where $V$ and $A$ are computed by the PyKVFinder [40] package. The dataset consists of 98k+ training samples and 100 testing samples for the atom number predictor.

**Baselines**    PAFlow is compared with the following baseline methods: **LiGAN** [22] is a CNN-Based conditional VAE generating voxelized atomic density. **AR** [3] and **Pocket2Mol** [4] are autoregressive generative models where atoms are sampled one by one. **TargetDiff** [8], **DecompDiff** [9], **IPDiff** [10], **TAGMol** [24] and **ALiDiff** [11] are diffusion-based models that generate 3D molecules in a non-autoregressive manner. **MolCRAFT** [15] generates molecules in the continuous parameter space using Bayesian Flow Network (BFN) framework. **FlowSBDD** [14] is based on the rectified flow model, another form of FM that transports prior to data along linear paths, with a novel bond distance loss.

**Evaluation metrics**    The generated molecules are evaluated from two aspects: **binding affinity** with the target protein and **molecular properties**. We employ the widely adopted AutoDock Vina[41] to estimate the mean and median values of affinity-based metrics (Vina Score, Vina Min, Vina Dock and High Affinity). Vina Score directly measures the binding affinity based on the generated poses;

Vina Min optimizes the pose through local minimization before estimation; Vina Dock performs re-docking to reflect the optimal binding affinity; High Affinity computes the percentage of generated molecules that bind better than the reference ligands per test protein. Following [3, 22], QED [42] (drug-likeness), SA [43] (synthesize accessibility), and diversity are adopted to evaluate critical molecular properties.

Table 1: Summary of binding affinity and molecular properties of reference molecules and molecules generated by PAFlow and other baselines. ($\uparrow$)/($\downarrow$) denotes a larger / smaller number is better. Top 2 results are highlighted with **bold text** and underlined text, respectively.

| Method | | Vina Score ($\downarrow$) | | Vina Min ($\downarrow$) | | Vina Dock ($\downarrow$) | | High Affinity ($\uparrow$) | | QED ($\uparrow$) | | SA ($\uparrow$) | | Diversity ($\uparrow$) | |
|---|---|---|---|---|---|---|---|---|---|---|---|---|---|---|---|
| | | Avg. | Med. | Avg. | Med. | Avg. | Med. | Avg. | Med. | Avg. | Med. | Avg. | Med. | Avg. | Med. |
| Ref | | -6.36 | -6.46 | -6.71 | -6.49 | -7.45 | -7.26 | - | - | 0.48 | 0.47 | 0.73 | 0.74 | - | - |
| CVAE | LiGAN | - | - | - | - | -6.33 | -6.20 | 21.1% | 11.1% | 0.39 | 0.39 | 0.59 | 0.57 | 0.66 | 0.67 |
| Auto-regressive | AR | -5.75 | -5.64 | -6.18 | -5.88 | -6.75 | -6.62 | 37.9% | 31.0% | 0.51 | 0.50 | 0.63 | 0.63 | 0.70 | 0.70 |
| | Pocket2Mol | -5.14 | -4.70 | -6.42 | -5.82 | -7.15 | -6.79 | 48.4% | 51.0% | **0.56** | **0.57** | **0.74** | **0.75** | 0.69 | 0.71 |
| Diffusion | TargetDiff | -5.47 | -6.30 | -6.64 | -6.83 | -7.80 | -7.91 | 58.1% | 59.1% | 0.48 | 0.48 | 0.58 | 0.58 | 0.72 | 0.71 |
| | DecompDiff | -5.67 | -6.04 | -7.04 | -7.09 | -8.39 | -8.43 | 64.4% | 71.0% | 0.45 | 0.43 | 0.61 | 0.60 | 0.68 | 0.68 |
| | IPDiff | -6.42 | -7.01 | -7.45 | -7.48 | -8.57 | -8.51 | 69.5% | 75.5% | 0.52 | 0.53 | 0.61 | 0.59 | 0.74 | 0.73 |
| | TAGMol | -7.02 | -7.77 | -7.95 | -8.07 | -8.59 | -8.69 | 69.8% | 76.4% | 0.55 | 0.56 | 0.56 | 0.56 | 0.69 | 0.70 |
| | ALiDiff | -7.07 | -7.95 | -8.09 | -8.17 | -8.90 | -8.81 | 73.4% | 81.4% | 0.50 | 0.50 | 0.57 | 0.56 | 0.73 | 0.71 |
| BFN | MolCRAFT | -6.59 | -7.04 | -7.27 | -7.26 | -7.92 | -8.01 | 59.1% | 62.6% | 0.50 | 0.51 | 0.69 | 0.68 | 0.73 | 0.73 |
| FM | FlowSBDD | -3.62 | -5.03 | -6.72 | -6.60 | -8.50 | -8.36 | 63.4% | 70.9% | 0.47 | 0.48 | 0.51 | 0.51 | **0.75** | **0.75** |
| | **Ours** | **-8.31** | **-8.92** | **-8.79** | **-8.96** | **-9.46** | **-9.49** | **80.8%** | **93.7%** | 0.49 | 0.50 | 0.57 | 0.57 | 0.71 | 0.70 |

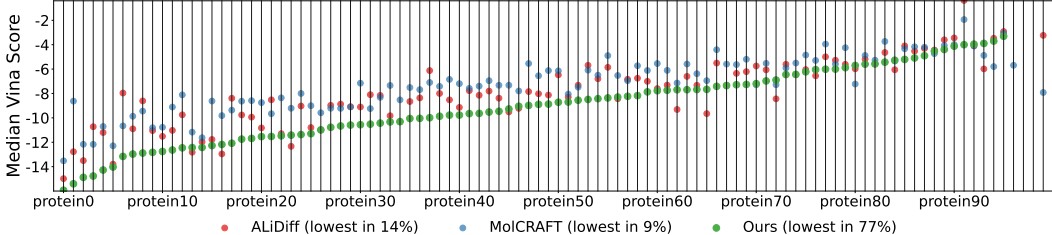

Figure 2: Median Vina energy for different generated molecules (ALiDiff, MolCRAFT, PAFlow) across 100 testing binding targets. The proteins are sorted by the median Vina energy of molecules generated from PAFlow.

## 5.2 Main Results

**Target Binding Affinity and Molecular Properties** We comprehensively evaluate the performance of PAFlow by comparing it against four types of SBDD methods: autoregressive, diffusion, BFN, and FM methods. 100 molecules are sampled for each test protein. As shown in Tab. 1, PAFlow significantly outperforms all baseline methods on all binding-related metrics, where the Vina score is computed without conformational optimization and serves as the most critical metric. Specifically, PAFlow surpasses the strong diffusion method ALiDiff by a large margin of 17.5%, 8.7%, and 6.3% on Avg. Vina Score, Vina Min, and Vina Dock, respectively. Furthermore, it also achieves superior performance over the novel BFN method MolCRAFT by 26.1%, 20.9%, and 19.4% on the same metrics. In terms of high-affinity binders, an average of 80.8% of the PAFlow molecules exhibit higher binding affinity than the reference ligands, significantly exceeding all other baselines. Notably, the performance of FlowSBDD—another FM-based model—is unremarkable in binding affinity due to the use of linear probability paths, inadequately capturing the complexity inherent to SBDD. In contrast, PAFlow designs appropriate probability paths for atomic coordinates and atom types, effectively generating molecules with high binding affinity. Fig. 2 presents the median Vina energy of all generated molecules for each binding pocket, comparing PAFlow with two SOTA methods ALiDiff and MolCRAFT. PAFlow achieves the highest binding affinity on 77% of the targets, significantly exceeding the other two methods.

PAFlow generates molecules with higher binding affinity in Tab. 1 while maintaining comparable QED, SA, and diversity to ALiDiff. These properties are not explicitly emphasized during generation,

as in real-world drug discovery they are typically used for rough filtering and are considered acceptable as long as they fall within a reasonable range. Based on the PAFlow framework, extending the molecule–protein interaction predictor to molecular properties holds promise for further improving these metrics. Fig. 3 presents examples of generated molecules along with their properties. The results show that the molecules generated by PAFlow exhibit favorable properties while maintain reasonable structures, suggesting strong potential as candidate ligands. We further conduct structural analysis experiments, with the results presented in Appendix E.7.

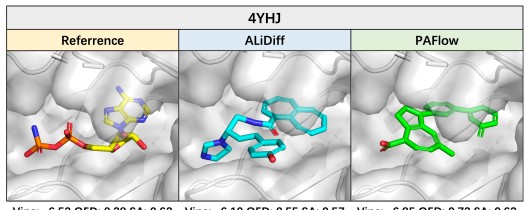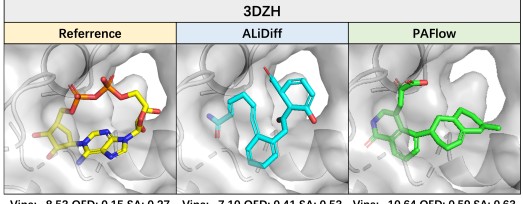

Figure 3: Visualizations of reference molecules and molecules generated by ALiDiff and PAFlow for protein pockets (4YHJ, 3DZH , 2Z3H and 2JJG). Vina Score, QED and SA are reported below.

**Sampling Efficiency**   We select the most efficient model from each category of baselines to compare sampling speed with PAFlow. Additionally, we report the sampling efficiency of PAFlow using a reduced number of steps T = 20. Fig. 4 shows the inference time for generating 100 molecules on average. While TargetDiff and Pocket2Mol require 3968s and 4009s respectively, PAFlow only takes 717s, achieving a $5.5\times$ speedup. Although PAFlow is slightly slower than MolCRAFT, it produces molecules with significantly higher binding affinity, which compensates for the additional time cost. Moreover, when using fewer steps T = 20, PAFlow becomes even faster than MolCRAFT (402s vs 287s) while still generating molecules with better binding performance (see Appendix E.2). This allows the sampling step size to be adjusted according to the trade-off between generation speed and molecule quality in different scenarios.

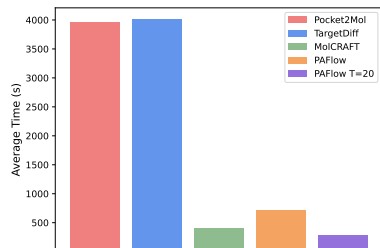

Figure 4: Average time required by different methods to generate 100 molecules for a target protein, with shorter times indicating higher sampling efficiency.

### 5.3   Ablation Studies

**Effectiveness of FM Framework**   As described in Sec. 4.1, PAFlow and TargetDiff share the same probability paths, but differ in their sampling strategies. To highlight the advantages of FM-based generation, we construct a variant denoted as PAFlow w/o PA, by removing prior guidance and replacing the atom count predictor with predefined sampling. This essentially corresponds to replacing the denoising strategy in TargetDiff with ODE-based sampling while keeping all other components unchanged. Evaluation results for the generated molecules are reported in Tab. 2. PAFlow w/o PA generally outperforms TargetDiff on the affinity-related metrics. Moreover, while maintaining the same SA score, it achieves even better QED and diversity than TargetDiff. These results demonstrate the superior generative performance of FM compared to diffusion-based denoising. Fig. 5 further visualizes the trajectories of atomic coordinates during generation, clearly showing that compared to diffusion the smoother probability path of FM could result in the better molecular quality.

**Impact of Prior Guidance**   To validate the effectiveness of guiding the vector field with the protein-ligand interaction predictor, we generate 1,000 molecules in all test proteins with PAFlow w/o P (which excludes prior guidance during sampling) and PAFlow. As demonstrated in Tab. 3, when guidance is applied, the generated molecules exhibit substantial improvements across all affinity-related metrics, with increases of 60.4%, 22.5%, and 13.0% in Avg. Vina Score, Vina Min, and Vina Dock, respectively. Improvements are also observed in SA and Diversity. Although QED shows a slight decrease, it remains within an acceptable range. Given the superior performance of the guidance strategy, extending the predictor to molecular properties holds great promise for further

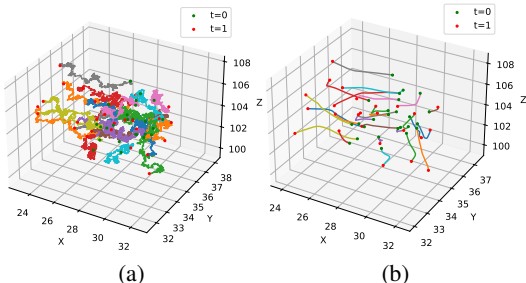

|        |        |
|:------:|:------:|
| (a)    | (b)    |

Figure 5: Atomic coordinate trajectories from $t = 0$ to $t = 1$ under different sampling strategies with the same probability paths. (a) shows the generation trajectory of TargetDiff, while (b) presents that of PAFlow w/o PA.

Table 2: Evaluation of generation results on TargetDiff and PAFlow w/o PA. PAFlow w/o PA is derived from TargetDiff by retaining all components unchanged except for replacing the sampling strategy with ODE-based sampling.

| Metrics | TargetDiff | | PAFlow w/o PA | |
|---------|------|------|------|------|
|         | Avg. | Med. | Avg. | Med. |
| Vina Score | **-5.47** | -6.30 | -5.13 | **-6.35** |
| Vina Min | -6.64 | -6.83 | **-6.76** | **-7.12** |
| Vina Dock | -7.80 | -7.91 | **-8.08** | **-8.18** |
| QED | 0.48 | 0.48 | **0.53** | **0.53** |
| SA | **0.58** | **0.58** | **0.58** | **0.58** |
| Diversity | 0.72 | 0.71 | **0.73** | **0.72** |

enhancing these metrics. Additionally, the extent of improvement is influenced by the guidance strength $\gamma$, with related results presented in the Appendix E.3.

Table 3: Effect of using the molecular-protein interaction predictor for guidance. *PAFlow w/o P* refers to the variant that does not incorporate prior knowledge to guide the vector field during the generation process.

| Methods | Vina Score (↓) | | Vina Min (↓) | | Vina Dock (↓) | | High Affinity (↑) | | QED (↑) | | SA (↑) | | Diversity (↑) | |
|---------|------|------|------|------|------|------|------|------|------|------|------|------|------|------|
|         | Avg. | Med. | Avg. | Med. | Avg. | Med. | Avg. | Med. | Avg. | Med. | Avg. | Med. | Avg. | Med. |
| PAFlow w/o P | -5.05 | -6.78 | -6.94 | -7.48 | -8.51 | -8.53 | 70.3% | 88.2% | **0.53** | **0.53** | 0.56 | 0.56 | **0.70** | 0.69 |
| PAFlow | **-8.10** | **-8.83** | **-8.50** | **-8.80** | **-9.62** | **-9.39** | **80.7%** | **100.0%** | 0.49 | 0.50 | **0.57** | **0.57** | **0.70** | **0.70** |

**Effect of Atom Number Predictor** To assess the impact of atom number predictor, we design three variant models for comparison. All modifications are based on PAFlow w/o P to eliminate the influence of prior-guided sampling: (1) predefined: atom numbers are sampled from the predefined distributions; (2) $\delta = 0$: atom numbers are directly predicted by the predictor without added noise; and (3) $\delta = 0.01$: Gaussian noise $\mathcal{N}(0, 0.01^2)$ is added to the predicted result. We generate 1,000 molecules for each setting on the test proteins, and the evaluation results are summarized in Tab. 7. Using the predictor yields substantial improvements in all affinity-related metrics compared to sampling from the predefined distributions. For molecular properties, QED and SA are improved while maintaining diversity. This demonstrates that molecules generated with predicted atom numbers exhibit more appropriate molecular sizes and thus perform better overall. Furthermore, adding a small Gaussian to the predicted result leads to additional gains in performance. This randomness offers robustness that deterministic outputs lack and helps compensate for minor prediction errors. Moreover, introducing noise allows model to explore a broader design space and potentially discover molecules with superior performance.

Table 4: Comparison of different strategies for determining the number of atoms in molecules generated by PAFlow w/o P.

| Methods | Vina Score (↓) | | Vina Min (↓) | | Vina Dock (↓) | | High Affinity (↑) | | QED (↑) | | SA (↑) | | Diversity (↑) | |
|---------|------|------|------|------|------|------|------|------|------|------|------|------|------|------|
|         | Avg. | Med. | Avg. | Med. | Avg. | Med. | Avg. | Med. | Avg. | Med. | Avg. | Med. | Avg. | Med. |
| predefined | -4.49 | -6.50 | -6.33 | -7.28 | -7.62 | -8.46 | 61.2% | 66.7% | 0.52 | 0.51 | 0.57 | 0.57 | **0.73** | 0.71 |
| $\delta = 0$ | -5.72 | -6.86 | -7.30 | -7.52 | -8.30 | -8.48 | 70.4% | **88.9%** | **0.55** | **0.56** | **0.58** | **0.58** | 0.72 | 0.71 |
| $\delta = 0.01$ | **-6.17** | **-7.03** | **-7.50** | **-7.74** | **-8.70** | **-8.61** | **72.8%** | **88.9%** | **0.55** | **0.56** | 0.57 | 0.57 | 0.71 | **0.71** |

# 6 Conclusion

In this work, we propose PAFlow, a flow matching-based SBDD method that separately models continuous atomic coordinates and discrete atom types using the VP path and a newly constructed CFM. By incorporating a protein-ligand interaction predictor, PAFlow effectively guides the vector field toward higher binding affinity. Additionally, an atom number predictor is introduced to determine the number of atoms in generated molecules, addressing the geometric incompatibility between generated molecules and target proteins and eliminating the dependence on prior knowledge from

reference ligands. Extensive experiments on the CrossDocked2020 benchmark demonstrate that PAFlow achieves state-of-the-art performance in terms of binding affinity, with up to -8.31 Avg. Vina score, while maintaining desirable molecular properties. In future work, we plan to extend the interaction predictor to molecular properties, enabling multi-objective optimization over both binding affinity and critical molecular properties for enhanced drug discovery potential.

## 7 Acknowledgements

This work was supported by the National Natural Science Foundation of China (grants No. 62172273), the Science and Technology Commission of Shanghai Municipality (grants No. 24510714300), and the Shanghai Municipal Science and Technology Major Project, China (Grant No. 2021SHZDZX0102).

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

# A Proofs

## A.1 Proof of Equivariant Generation

An essential requirement for generated molecules is translational and rotational equivariance to the ligand–protein complex. Since atom types are always invariant to SE(3)-transformation, we only need to consider the transformation behavior of atomic coordinates. As the probability path used during training is identical to that of TargetDiff [8]—and the path of TargetDiff is invariant by design—we only have to focus on the generation process. Let $T_g$ denote the group of SE-(3) transformation, e.g. $T_g(\mathbf{x}) = \mathbf{R}\mathbf{x} + \mathbf{b}$, where $\mathbf{R} \in \mathbb{R}^{3\times3}$ is the rotation matrix and $\mathbf{b} \in \mathbb{R}^3$ is the translation vector.

First, we move the complex to achieve zero Center-of-Mass (CoM) on protein positions by the following linear transformation:

$$[\bar{\mathbf{x}}_M, \bar{\mathbf{x}}_P] = Q(\mathbf{x}_M, \mathbf{x}_P), \qquad where \quad Q = I_3 \otimes \begin{pmatrix} I_{N_M} & -\frac{1}{N_P}\mathbf{1}_{N_M}\mathbf{1}_{N_P}^T \\ \mathbf{0} & I_{N_P} - \frac{1}{N_P}\mathbf{1}_{N_P}\mathbf{1}_{N_P}^T \end{pmatrix}. \tag{17}$$

Then flow can be initialized from standard Gaussian, and during the generation process $\bar{\mathbf{x}}_0$ can be sampled from a standard Gaussian distribution. Meanwhile, for any $T_g$ applied to the protein and molecule, we have $T_g(\bar{\mathbf{x}}_M, \bar{\mathbf{x}}_P) = \mathbf{R}(\bar{\mathbf{x}}_M, \bar{\mathbf{x}}_P)$. Thus $\bar{\mathbf{x}}_M$ and $\bar{\mathbf{x}}_P$ inherently satisfy translational invariance by definition. For brevity, we denote $\bar{\mathbf{x}}_M, \bar{\mathbf{x}}_P$ by $\mathbf{x}_M, \mathbf{x}_P$ in the following.

Given that $\phi_\theta$ is an SE(3)-equivariant GNN, it follows from Eq. 9 that $\hat{\mathbf{x}}_1$ is SE-(3) equivariant to $\mathbf{x}_t$ and $\mathbf{x}_P$. At $t = 0$, based on Eq. 5 and 8 we obtain

$$\mathbf{x}_{\Delta t} = \mathbf{x}_0 + v_\theta \Delta t$$
$$= \mathbf{x}_0 + \frac{(\sqrt{\bar{\alpha}_{1-t}})'}{1 - \bar{\alpha}_{1-t}}(\sqrt{\bar{\alpha}_{1-t}}\mathbf{x}_0 - \hat{\mathbf{x}}_1). \tag{18}$$

We can prove that $\mathbf{x}_{\Delta t}$ is SE(3)-equivariant to $\mathbf{x}_0$ and $\mathbf{x}_P$ as follows

$$T_g(\mathbf{x}_{\Delta t}(\mathbf{x}_0, \mathbf{x}_P)) = T_g(\mathbf{x}_0) + \frac{(\sqrt{\bar{\alpha}_{1-t}})'}{1 - \bar{\alpha}_{1-t}}(\sqrt{\bar{\alpha}_{1-t}}T_g(\mathbf{x}_0) - T_g(\hat{\mathbf{x}}_1))$$
$$= \mathbf{R}\mathbf{x}_0 + \mathbf{b} + \frac{(\sqrt{\bar{\alpha}_{1-t}})'}{1 - \bar{\alpha}_{1-t}}(\sqrt{\bar{\alpha}_{1-t}}\mathbf{R}\mathbf{x}_0 + \sqrt{\bar{\alpha}_{1-t}}\mathbf{b} - \mathbf{R}\hat{\mathbf{x}}_1 - \mathbf{b})$$
$$= \mathbf{R}\mathbf{x}_0 + \frac{(\sqrt{\bar{\alpha}_{1-t}})'}{1 - \bar{\alpha}_{1-t}}\mathbf{R}(\sqrt{\bar{\alpha}_{1-t}}\mathbf{x}_0 - \hat{\mathbf{x}}_1)\Delta t + \tilde{\mathbf{b}}$$
$$= \mathbf{x}_{\Delta t}(\mathbf{R}(\mathbf{x}_0, \mathbf{x}_P)) + \tilde{\mathbf{b}}, \tag{19}$$

where $\tilde{\mathbf{b}} = [1 - (\sqrt{\bar{\alpha}_{1-t}})'(1 + \sqrt{\bar{\alpha}_{1-t}})\Delta t]\mathbf{b}$. Since translation invariance is achieved by moving CoM of the protein atoms to zero, $\tilde{\mathbf{b}}$ can be omitted thereby $T_g(\mathbf{x}_{\Delta t}(\mathbf{x}_0, \mathbf{x}_P)) = \mathbf{x}_{\Delta t}(\mathbf{R}(\mathbf{x}_0, \mathbf{x}_P)) = \mathbf{x}_{\Delta t}(T_g(\mathbf{x}_0, \mathbf{x}_P))$. Following the same reasoning, it can be derived that $\mathbf{x}_{t+\Delta t}$ is SE(3)-equivariant to $\mathbf{x}_t$ and $\mathbf{x}_P$. By mathematical induction, we conclude that $\mathbf{x}_1$ is SE(3)-equivariant to $\mathbf{x}_0$ and $\mathbf{x}_P$. Consequently, the entire generation process is SE(3)-equivariant with respect to $\mathbf{x}_0$ and $\mathbf{x}_P$.

## A.2 Vector Field of Type Flow

*Theorem 3* in [16] propose that for a flow of the form $\psi_t(x) = r_t(x_1)x + s_t(x_1)$, the corresponding vector field is given by:

$$u_t(x|x_1) = \frac{r_t'(x_1)}{r_t(x_1)}(x - s_t(x_1)) + s_t'(x_1), \tag{20}$$

where $r_t(x_1)$ and $s_t(x_1)$ can be arbitrary differentiable function satisfying the desired boundary conditions. Let $s_t = \bar{\alpha}_{1-t}\mathbf{a}_1$, $r_t = 1 - \bar{\alpha}_{1-t}$, then Eq. 20 is rewritten as

$$u_t(x|x_1) = -\frac{\bar{\alpha}_{1-t}'}{1 - \bar{\alpha}_{1-t}}(x - \bar{\alpha}_{1-t}x_1) + \bar{\alpha}_{1-t}'x_1. \tag{21}$$

For atomic types we have $\mathbf{c}(\mathbf{a}, \mathbf{a}_1) = \bar{\alpha}_{1-t}\mathbf{a}_1 + (1 - \bar{\alpha}_{1-t})\mathbf{a}_0$, which can be substituted into Eq. 21 to obtain:

$$
\begin{aligned}
u_t(\mathbf{c}(\mathbf{a}, \mathbf{a}_1)|\mathbf{a}_1) &= -\frac{\bar{\alpha}'_{1-t}}{1 - \bar{\alpha}_{1-t}}(\bar{\alpha}_{1-t}\mathbf{a}_1 + (1 - \bar{\alpha}_{1-t})\mathbf{a}_0 - \bar{\alpha}_{1-t}\mathbf{a}_1) + \bar{\alpha}'_{1-t}\mathbf{a}_1 \\
&= -\frac{\bar{\alpha}'_{1-t}}{1 - \bar{\alpha}_{1-t}}(1 - \bar{\alpha}_{1-t})\mathbf{a}_0 + \bar{\alpha}'_{1-t}\mathbf{a}_1 \\
&= \bar{\alpha}'_{1-t}(\mathbf{a}_1 - \mathbf{a}_0).
\end{aligned}
\tag{22}
$$

### A.3  Prior Guidance of Vector Field

According to *Lemma 1* in [37], the guided vector field for a Gaussian Path $p_t(x|x_1) = \mathcal{N}(x|\mu_t x_1, \sigma_t^2 I)$ can be expressed as:

$$
u_t(x|y) = a_t x_t + b_t \nabla \log p(x_t|y),
\tag{23}
$$

$$
where \quad a_t = \frac{\mu'_t}{\mu_t}, \quad b_t = (\mu'_t \sigma_t - \mu_t \sigma'_t)\frac{\sigma_t}{\mu_t}.
\tag{24}
$$

The probability path for atomic coordinates follows the Gaussian distribution with $\mu_t = \sqrt{\bar{\alpha}_{1-t}}\mathbf{x}_1$ and $\sigma_t = \sqrt{1 - \bar{\alpha}_{1-t}}$. Substituting into Eq. 24 yields:

$$
a_t = \frac{(\sqrt{\bar{\alpha}_{1-t}}\mathbf{x}_1)'}{\sqrt{\bar{\alpha}_{1-t}}\mathbf{x}_1} = \frac{\bar{\alpha}'_{1-t}}{2\bar{\alpha}_{1-t}},
\tag{25}
$$

$$
\begin{aligned}
b_t &= \left[\frac{\bar{\alpha}'_{1-t}\sqrt{1 - \bar{\alpha}_{1-t}}}{2\sqrt{\bar{\alpha}_{1-t}}}x_1 + \frac{\bar{\alpha}'_{1-t}\sqrt{\bar{\alpha}_{1-t}}}{2\sqrt{1 - \bar{\alpha}_{1-t}}}x_1\right] \frac{\sqrt{1 - \bar{\alpha}_{1-t}}}{x_1\sqrt{\bar{\alpha}_{1-t}}} \\
&= \frac{\bar{\alpha}'_{1-t}}{2}\left(\frac{1 - \bar{\alpha}_{1-t}}{\bar{\alpha}_{1-t}} + 1\right) \\
&= \frac{\bar{\alpha}'_{1-t}}{2\bar{\alpha}_{1-t}}.
\end{aligned}
\tag{26}
$$

Thus we have $a_t = b_t = \frac{\bar{\alpha}'_{1-t}}{2\bar{\alpha}_{1-t}}$. Substituting this into Eq. 23 gives the guided vector field for atomic coordinates as follows:

$$
\tilde{v}_\theta^x(\mathbf{m}_t, \mathbf{p}, y, t) = \frac{\bar{\alpha}'_{1-t}}{2\bar{\alpha}_{1-t}}(\mathbf{x}_t + \nabla \log p(\mathbf{x}_t|y)).
\tag{27}
$$

According to the Bayes' rule, log-probability function can be decomposed as:

$$
\begin{aligned}
\nabla \log p(\mathbf{x}_t|y) &= \nabla(\log p(\mathbf{x}_t) + \log p(y|\mathbf{x}_t) - \log p(y)) \\
&= \nabla \log p(\mathbf{x}_t) + \nabla \log p(y|\mathbf{x}_t),
\end{aligned}
\tag{28}
$$

where $\nabla \log p(y|\mathbf{x}_t)$ can be calculated using predictor with Eq. 15. Then we can reformulate Eq. 27 as:

$$
\begin{aligned}
\tilde{v}_\theta^x(\mathbf{m}_t, \mathbf{p}, y, t) &= \frac{\bar{\alpha}'_{1-t}}{2\bar{\alpha}_{1-t}}(\mathbf{x}_t + \nabla \log p(\mathbf{x}_t)) + \frac{\bar{\alpha}'_{1-t}}{2\bar{\alpha}_{1-t}}\nabla \log p(y|\mathbf{x}_t) \\
&= v_\theta^x(\mathbf{m}_t, \mathbf{p}, t) + \frac{\bar{\alpha}'_{1-t}}{2\bar{\alpha}_{1-t}}\nabla \log p(y|\mathbf{x}_t).
\end{aligned}
\tag{29}
$$

Moreover, scaling the predictor gradient by a constant factor $\gamma > 1$ is necessary to enhance the alignment of the generated samples with the desired condition. Similar to [44], we know that $\gamma \cdot \nabla \log p(y|\mathbf{x}_t) = \nabla \log \frac{1}{Z}p(y|\mathbf{x}_t)^\gamma$, where $Z$ is an arbitrary constant. Consequently, the conditioning procedure remains theoretically grounded in a re-normalized predictor distribution proportional to $p(y|\mathbf{x}_t)^\gamma$. Higher probabilities are emphasized by the exponent when $\gamma > 1$, leading to a distribution more peaked than the original $p(y|\mathbf{x}_t)$. Thus increasing the gradient scale focuses more on the modes of the predictor, which promotes the generation of molecules with higher binding affinity.

### A.4 Proof of Noise Injection Effectiveness

As introduced in Sec. 4.3, the predicted normalized number of atoms $\hat{n}_M$ in the generated molecule is obtained via the atom number predictor. For simplicity, we denote it as $n$ in the following. Using $n$ directly corresponds to deterministic sampling, i.e., $\tilde{n} = n$, whereas adding a small Gaussian noise corresponds to stochastic sampling with $\tilde{n} \sim \mathcal{N}(n, \delta^2)$. Let $f(n)$ denote the binding affinity as a function of the atom number $n$. The objective is to maximize the expected binding affinity, defined as $R = \mathbb{E}_{\tilde{n}}[f(\tilde{n})]$.

Under deterministic sampling, we have

$$R = \mathbb{E}_n[f(n)] = f(n). \tag{30}$$

When Gaussian noise is injected,

$$R = \mathbb{E}_{\tilde{n} \sim \mathcal{N}(n, \delta^2)}[f(\tilde{n})]. \tag{31}$$

The second-order Taylor expansion of $f(\tilde{n})$ at $n = \tilde{n}$ is given by:

$$f(\tilde{n}) \approx f(n) + f'(n)(\tilde{n} - n) + \frac{1}{2}f''(n)(\tilde{n} - n)^2. \tag{32}$$

Since $\tilde{n} \sim \mathcal{N}(n, \delta^2)$, the expectation can be rewritten as

$$\mathbb{E}_{\tilde{n}}[f(\tilde{n})] \approx f(n) + \frac{1}{2}f''(n)\delta^2. \tag{33}$$

As a result, the expected affinity increases over the deterministic value $f(n)$ if $f''(n) > 0$, i.e., if the function is locally convex around $n$. In practice, we observe that adding a small amount of noise (e.g., $\delta = 0.01$) consistently improves the binding affinity of the generated molecules (see Tab. 7). This implies that $n$ lies near a local minimum of the affinity function, and noise injection effectively allows exploration of nearby data points that lead to better binding performance.

## B   Training Objective

The training objectives for atomic coordinates $\mathbf{x}$, atomic types $\mathbf{a}$, protein-ligand interactions $y$, and atom numbers $n_M$ are defined as follows:

$$\mathcal{L}_x = ||\mathbf{x}_1 - \hat{\mathbf{x}}_1||^2 \qquad \mathcal{L}_a = \sum_k c(\mathbf{a}_t, \mathbf{a}_1)_k \log \frac{c(\mathbf{a}_t, \mathbf{a}_1)_k}{c(\mathbf{a}_t, \hat{\mathbf{a}}_1)_k} \tag{34}$$

$$\mathcal{L}_y = ||y - \hat{y}||^2 \qquad \mathcal{L}_n = ||n_M - \hat{n}_M||^2, \tag{35}$$

where the loss for $\mathbf{a}$ is KL-divergence of categorical distributions, while the others are optimized using Mean Squared Error. KGDiff demonstrates that jointly training $\mathbf{x}$, $\mathbf{a}$, and $y$ is more effective than training separately. Following this strategy, we adopt a joint training scheme with the combined loss defined as:

$$\mathcal{L} = \mathcal{L}_x + \lambda \mathcal{L}_a + \omega \mathcal{L}_y, \tag{36}$$

where $\lambda$ and $\omega$ are scaling factors. The atom number predictor is trained independently.

## C   Algorithm

The training and sampling procedure of PAFlow are summarized below.

---

**Algorithm 1** Training Procedure of PAFlow

---

**Input:** Protein-ligand complex $\{\mathbf{p}, \mathbf{m}, y\}_{i=1}^N$, neural network $\phi_\theta$, atom type loss weight $\lambda$ and predictor loss weight $\omega$

1: **while** $\phi_\theta$ not converage **do**
2:     Sample $t \sim \mathcal{U}(0, 1)$
3:     Shift the complex to make the CoM of protein atoms zero
4:     Obtain $\mathbf{x}_t$ and $\mathbf{a}_t$:
5:         $\mathbf{x}_t = \sqrt{\bar{\alpha}_{1-t}}\mathbf{x}_1 + \sqrt{(1 - \bar{\alpha}_{1-t})}\varepsilon$, where $\varepsilon \in \mathcal{N}(0, \mathbf{I})$
6:         $\log \mathbf{c} = \log(\bar{\alpha}_{1-t}\mathbf{a}_1 + (1 - \bar{\alpha}_{1-t})/K)$
7:         $\mathbf{a}_t = one\_hot(\text{argmax}_i[g_i + \log c_i])$, where $g \sim \text{Gumbel}(0, 1)$
8:     Predict $[\hat{\mathbf{x}}_1, \hat{\mathbf{a}}_1, \hat{y}]$ with $\phi_\theta$: $[\hat{\mathbf{x}}_1, \hat{\mathbf{a}}_1, \hat{y}] = \phi_\theta([\mathbf{x}_t, \mathbf{a}_t], t, \mathbf{p})$
9:     Compute the loss function using Eq. 36:
10:         $\mathcal{L} = \| \mathbf{x}_1 - \hat{\mathbf{x}}_1 \|^2 + \lambda\text{KL}(\mathbf{c}(\mathbf{a}_t, \mathbf{a}_1)\|\mathbf{c}(\mathbf{a}_t, \hat{\mathbf{a}}_1)) + \omega \| y - \hat{y}\|^2$
11:     Update $\theta$ by minimizing $\mathcal{L}$
12: **end while**

---

---

**Algorithm 2** Sampling Procedure of PAFlow

---

**Input:** Protein pocket $\mathbf{p}$, the learned model $\phi_\theta$, pretrained atom number predictor $\varphi_\epsilon$, scale factor $\gamma$, standard deviation of Gaussian $\delta$, total number of sampling steps $T$
**Output:** Generated ligand molecule $\mathbf{m}$

1: Obtain the number of atoms in $\mathbf{m}$ using Eq. 16
2: Move the CoM of protein atoms to zero
3: Initialize ligand atom coordinates $\mathbf{x}_0$ and atom types $\mathbf{a}_0$
4: $steps \leftarrow 0, t \leftarrow 0, \Delta t = 1/T$
5: **while** $steps \leq T - 1$ **do**
6:     Predict $[\hat{\mathbf{x}}_1, \hat{\mathbf{a}}_1, \hat{y}]$ with $\phi_\theta$: $[\hat{\mathbf{x}}_1, \hat{\mathbf{a}}_1, \hat{y}] = \phi_\theta([\mathbf{x}_t, \mathbf{a}_t], t, \mathbf{p})$
7:     Compute vector fields:
8:         $\tilde{v}_\theta^x = \frac{(\sqrt{\bar{\alpha}_{1-t}})'}{1 - \bar{\alpha}_{1-t}}(\sqrt{\bar{\alpha}_{1-t}}\mathbf{x}_t - \hat{\mathbf{x}}_1) + \gamma\frac{\bar{\alpha}'_{1-t}}{2\bar{\alpha}_{1-t}}\nabla \log p(y = 1|\mathbf{m}_t)$
9:         $v_\theta^a = \bar{\alpha}'_{1-t}(\hat{\mathbf{a}}_1 - \mathbf{a}_0)$
10:     Update $\mathbf{x}_{t+\Delta t}$ and $\mathbf{a}_{t+\Delta t}$:
11:         $\mathbf{x}_{t+\Delta t} = \mathbf{x}_t + \tilde{v}_\theta^x \Delta t$
12:         $\mathbf{c}(\mathbf{a}_{t+\Delta t}, \hat{\mathbf{a}}_1) = \mathbf{c}(\mathbf{a}_t, \hat{\mathbf{a}}_1) + v_\theta^a \Delta t$
13:         Sample $\mathbf{a}_t$ from $\mathcal{C}(\mathbf{a}_{t+\Delta t}|\mathbf{c}(\mathbf{a}_{t+\Delta t}, \hat{\mathbf{a}}_1))$
14: **end while**

---

## D   Implementation Details

**Data**   Following [8], the protein atom features include a one-hot indicator of the element type (H, C, N, O, S, Se), a one-hot vector indicating the amino acid type (20 dimension), and a one-dim flag indicating whether the atom belongs to the protein backbone. Ligand atom types are also represented as one-hot vectors that encode the element type (C, N, O, F, P, S, Cl) along with aromatic information. Two separate single-layer MLPs are utilized to embed the protein and ligand features into a 128-dimensional latent space.

The protein-ligand complex is adaptively represented as a $k$-nearest neighbors (knn) graph at the $l$-th layer, constructed using the known protein atom coordinates and the current ligand atom coordinates generated by the $l - 1$-th layer. $k = 32$ is set in experiments. The edge features are obtained as the outer product between distance embeddings and bond type, while distances are expanded using radial basis functions centered at 20 points distributed between 0 Å and 10 Å. Bond types are encoded as a 4-dimensional one-hot vector indicating whether the connection is protein–protein, ligand–ligand, protein-to-ligand, or ligand-to-protein.

**Model Information**  The SE(3)-equivariant network $\phi_\theta$ consists of 9 equivariant layers, where each layer is implemented as a transformer with $\text{hidden\_dim} = 128$ and $\text{n\_head} = 16$. The key/value embeddings and attention scores are generated through 2-layer MLPs with ReLU activation and Layer Normalization. The protein-ligand interaction predictor employs a 2-layer MLP with ShiftedSoftplus activation. For the atom number predictor $\varphi_\epsilon$, we adopt a 4-layer MLP with hidden dimensions of 128, 256, and 128. Each hidden layer is followed by Batch Normalization and ShiftedSoftplus activation, along with Dropout for regularization. The final output is produced by a linear projection to a scalar. We apply a sigmoid $\beta$ schedule with $\beta_1 = 1\mathrm{e}{-7}$ and $\beta_0 = 2\mathrm{e}{-3}$ for atom coordinates, and a cosine $\beta$ schedule proposed by [45] with $s = 0.01$ for atom types.

**Training Details**  When training the SE(3)-Equivariant network, the Adam optimizer is employed to speed up convergence with $\text{init\_learning\_rate} = 5\mathrm{e}-4$, $\text{betas} = (0.95, 0.999)$, $\text{batch\_size} = 4$ and $\text{clip\_gradient\_norm} = 8$. The learning rate is scheduled to decay exponentially with a decay factor of $0.95$, and we set the minimum learning rate to be $1\mathrm{e}-6$. We decay the learning rate if the validation loss is not improved for 15 consecutive evaluations. Following [23], we set $\lambda$, the weight corresponding to the atom feature term in loss function, is 100, and $\omega$, the weight corresponding to the expert network, is 1. When generating, the scaling factor $\gamma$ is 350 and the number of sampling steps is set to 50.

During the training of the atom number predictor, we use the Adam optimizer with $\text{init\_learning\_rate} = 5\mathrm{e}-4$, $\text{batch\_size} = 256$, and $\text{betas} = (0.95, 0.999)$. The learning rate follows an exponential decay schedule with a decay factor of $0.8$, and a minimum learning rate of $1\mathrm{e}-5$ is enforced. The learning rate is decayed when the validation loss does not improve for 5 consecutive evaluations. All models are trained on one NVIDIA A100 GPU (40 GB).

Table 5: Overview of the properties of the reference ligands and the molecules generated by different methods on the **Binding MOAD** dataset. ($\uparrow$)/($\downarrow$) denotes a larger / smaller number is better. Top 2 results are highlighted with **bold text** and underlined text, respectively.

| Method | Vina Score ($\downarrow$) | | Vina Min ($\downarrow$) | | Vina Dock ($\downarrow$) | | High Affinity ($\uparrow$) | | QED ($\uparrow$) | | SA ($\uparrow$) | | Diversity ($\uparrow$) | |
|---|---|---|---|---|---|---|---|---|---|---|---|---|---|---|
| | Avg. | Med. | Avg. | Med. | Avg. | Med. | Avg. | Med. | Avg. | Med. | Avg. | Med. | Avg. | Med. |
| Ref | -6.32 | -5.82 | -7.22 | -6.65 | -7.95 | -7.67 | - | - | 0.65 | 0.63 | 0.35 | 0.34 | - | - |
| Pocket2Mol | -4.85 | -4.55 | -5.95 | -5.54 | -6.66 | -6.31 | 46.2% | 33.5% | **0.62** | **0.62** | **0.83** | **0.84** | **0.83** | **0.83** |
| TargetDiff | -5.60 | -5.49 | -6.48 | -6.15 | -7.47 | -7.20 | 62.3% | 63.0% | 0.54 | 0.55 | 0.62 | 0.62 | 0.75 | 0.76 |
| IPDiff | -6.64 | -6.80 | -7.50 | -7.31 | -8.57 | -8.25 | 77.6% | 86.4% | 0.53 | 0.53 | 0.58 | 0.58 | 0.74 | 0.74 |
| MolCRAFT | -7.33 | -6.90 | -7.54 | -7.10 | -7.90 | -7.50 | 65.3% | 77.4% | 0.55 | 0.56 | 0.70 | 0.69 | 0.74 | 0.76 |
| PAFlow | **-9.12** | **-8.87** | **-9.18** | **-8.78** | **-9.69** | **-9.19** | **83.4%** | **96.6%** | 0.48 | 0.46 | 0.58 | 0.58 | 0.70 | 0.70 |

# E  More Experimental Results

## E.1  Results on Binding MOAD

To investigate the performance of PAFlow across different datasets, we evaluate it on the Binding MOAD dataset [46] without additional retraining, which is a commonly used benchmark comprising experimentally determined protein–ligand complexes. We apply filtering based on the proteins' enzyme commission numbers [47] and exclude entries that could not be processed, resulting in 100 protein-ligand pairs for testing following previous work [48]. PAFlow and the baseline methods generate 100 molecules for each protein pocket, with the evaluation results summarized in Table 5. PAFlow achieves the best performance across all binding-related metrics, outperforming the second-best method by 24.4%, 21.8%, and 13.1% on Avg. Vina Score, Vina Min, and Vina Dock, respectively. Although there is a slight decline in molecular properties, they remain within a reasonable range. It is worth noting that PAFlow is not trained on the Binding MOAD dataset, indicating its strong generalization capability. Furthermore, it is reasonable to expect that performance could be further improved by training on this dataset.

## E.2 Effect of Sampling Steps

We conduct an ablation study on the sampling steps of PAFlow by generating 10 molecules each for 100 test proteins using sampling strategies with different step (20, 50, 80, 200). The resulting Vina Score, Vina Dock, QED, and SA curves are shown in Fig. 6. It can be observed that PAFlow achieves very competitive results on affinity-related metrics even with only 20 steps, outperforming all baseline methods listed in Tab. 1. In contrast, diffusion-based models typically require 1000 sampling steps, indicating that PAFlow significantly improves sampling efficiency. Moreover, increasing the sampling step further enhances performance across all metrics, since more sampling steps allows model to more closely approximate the probability flow, leading to a smoother and more accurate trajectory toward the target distribution. Considering the trade-off between sampling efficiency and overall molecular quality, the number of sampling steps is set to 50 for PAFlow.

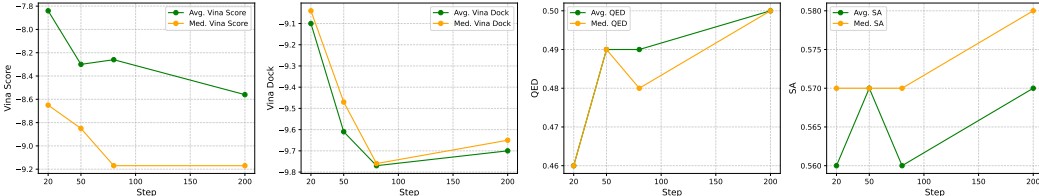

Figure 6: Ablation study on sampling steps. The performance of Vina Score, Vina Dock, QED, and SA under different numbers of sampling steps (20, 50, 80, and 200) is reported. Lower Vina Score and Vina Dock, as well as higher QED and SA, indicate better results.

## E.3 Effect of Scaling Factor in Guidance

The scaling factor $\gamma$, introduced in Sec. 4.2, is utilized to control the strength of guidance applied to the vector field. We generate 10 molecules each for 100 test target proteins using PAFlow with different $\gamma$ values (0, 10, 50, 150, 250, 350, 400). The results are presented in Fig. 7. A larger $\gamma$ corresponds to stronger guidance, which leads to higher binding affinities for the generated molecules. However, a clear trade-off between binding affinity and QED can be observed, where as binding affinity improves, QED tends to decrease. This is consistent with the discussion in Sec. A.3 that larger gradient scales concentrate more on the modes of the predictor, potentially at the expense of other molecular properties. Considering both binding affinity and molecular properties, we select $\gamma = 350$ for PAFlow. Additionally, $\gamma$ can be tuned during molecule generation to achieve different trade-offs between binding affinity and molecular properties, allowing adaptation to various drug design scenarios.

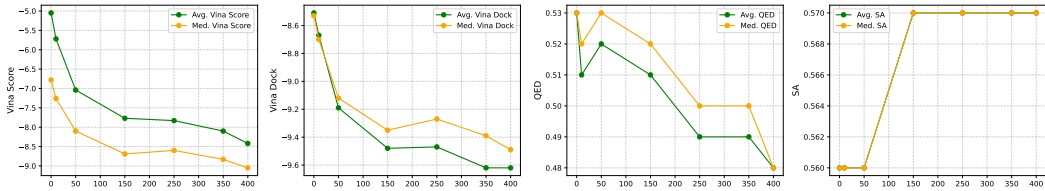

Figure 7: Ablation study of PAFlow under different guidance scaling factor $\gamma$. The performance on Vina Score, Vina Dock, QED, and SA is reported. In the SA plot, only one curve is shown because the average and median SA are identical.

## E.4 Effect of the Gaussian standard deviation

To investigate the impact of different $\delta$ on molecular quality, we generated 10 molecules for each of 100 test proteins, where $\delta$ is the standard deviation of small Gaussian noise added to the output of the atom count predictor. Fig. 8 reports the averages of Vina Score, QED, SA, and the proportion of complete molecules under $\delta=$(0, 0.02, 0.04, 0.06, 0.08, 0.1). When $\delta$ is small, both binding affinity and the proportion of complete molecules improve while maintaining molecular properties. This

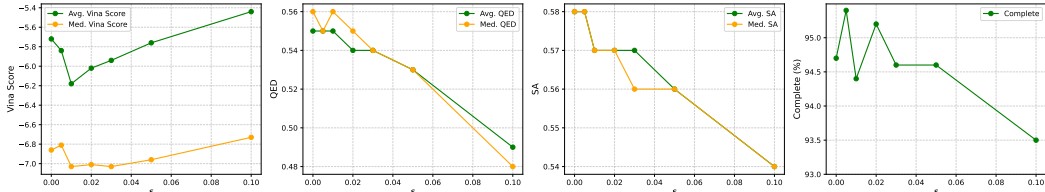

Figure 8: Ablation study using Gaussian noise with different standard deviations added to the predicted atom number. Vina Score, QED, SA, and the percentage of complete molecules are reported.

is because the added randomness introduces robustness that deterministic outputs lack, helping to compensate for minor prediction errors. However, when the standard deviation is larger, the noise injected overly disturbs the predicted values, leading to a decline in molecular quality. For PAFlow, we chose $\delta = 0.01$.

### E.5 Factors Influencing the Number of Atoms in Ligands

To validate the effectiveness of the protein features used for predicting the atom numbers in generated molecules (including the number of pocket atoms $N_P$, binding site volume $V$, binding site surface area $A$, and space size $S$), we conduct a study on their relationships using the training dataset. $V$ and $A$ are computed using pyKVFinder, while $S$ is the median value of the top 10 largest pairwise distances among protein atoms. Notably, existing non-autoregressive methods commonly use $S$ alone to determine the atom number. The relationships between these features and $N_M$ are shown in Fig. 9. All features exhibit significant correlations $(p < 0.05)$ with $N_M$, supporting the reasonableness of

Table 6: Comparison of evaluation results using only binding site volume versus using all four features.

| Metrics | Volume | All |
|---------|--------|-----|
| Test Loss (e-3) | 9.03 | **3.80** |
| Vina Score | -4.88 | **-5.72** |
| QED | **0.59** | 0.55 |
| SA | 0.56 | **0.58** |
| Diversity | 0.69 | **0.72** |
| Complete | 94.0% | **94.7%** |

using them for atom number prediction. Additionally, we compare the performance of using only $V$ (denoted as Volume) versus using all four features (denoted as All), as reported in Tab. 6. The test loss refers to the loss of the trained predictor evaluated on the test set. Results show that incorporating all features leads to a lower test loss and better molecule quality, indicating that providing richer protein information allows the predictor to more accurately infer the appropriate $N_M$.

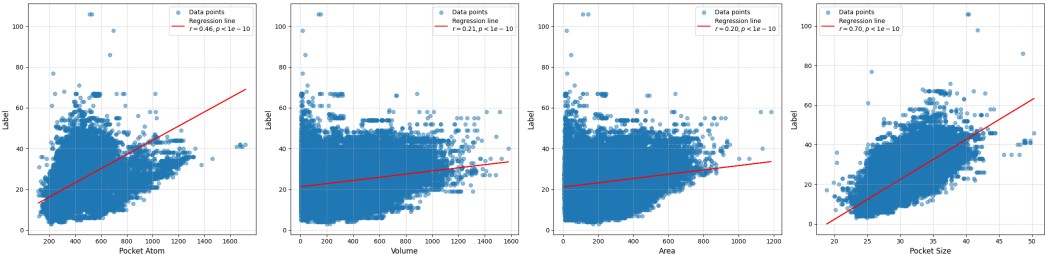

Figure 9: Relationships between ligand atom number and pocket atom number, binding site volume, surface area, and space size in the training set. Red lines show regression fits. Correlation coefficients and p-values (all $< 1e - 10$) are reported.

### E.6 Validation for Atom Number Predictor

Prior research [49] has established a strong positive correlation between ligand size (i.e., number of atoms) and binding affinity. However, the performance gain in our method does not stem from simply generating molecules with more atoms. Instead, it results from generating molecules whose atom counts are closer to the reference ligand.

To more intuitively demonstrate the effectiveness of our predictor, we compare it with the predefined sampling by generating 1,000 atom numbers for each of three randomly selected test proteins. For each protein, we define 20% and 30% intervals around the reference atom number and compute the proportion of generated samples falling within these ranges. As shown in Table 7, most atom numbers predicted by our model fall within the 20% range of the reference ligand's atom count, without producing significantly larger molecules. In contrast, the traditional method—which samples atom numbers from predefined distributions—yields a substantial portion of results outside the 30% range. Such deviation increases the likelihood of generating molecules that are either too large or too small to fit well within the binding pocket, thereby negatively affecting the protein–ligand interaction.

Table 7: Comparison between predefined sampling and our predictor on atom number. For each of the three randomly selected test proteins, 1,000 atom numbers are generated.

| PDB ID | | <70% | [70%, 80%) | [80%, 100%) | [100%, 120%) | [120%, 130%) | >130% |
|---|---|---|---|---|---|---|---|
| 4YHJ | Predefined | 4.7% | 10.9% | 26.2% | 34.2% | 12.5% | 11.5% |
| | Predict | 0 | 0 | 64.0% | 36.0% | 0 | 0 |
| 2E24 | Predefined | 7.4% | 7.8% | 17.1% | 17.4% | 13.7% | 36.6% |
| | Predict | 0 | 0 | 2.9% | 81.4% | 15.5% | 0.2% |
| 3NFB | Predefined | 25.2% | 11.1% | 29.8% | 23.1% | 5.8% | 5.0% |
| | Predict | 0 | 0.4% | 97.2% | 2.4% | 0 | 0 |

### E.7 Structure Analysis

To evaluate whether PAFlow can generate valid molecular conformations, a series of structure-related experiments are conducted. We report the average Jensen-Shannon divergence (JSD) between the bond length, bond angle, and torsion angle distributions of the generated and reference molecules, where a lower JSD indicates a distribution closer to that of real structures. In addition, we provide the ring size distribution of generated molecules. Together, these evaluations comprehensively assess the substructure stability. To evaluate global structural stability, the strain energy (SE) and steric clash (Clash) statistics are also reported, which are calculated by PoseCheck [50]. As shown in Table 8-10, PAFlow exhibits comparable performance to existing methods across both local and global structural metrics. Although classifier guidance is known to sometimes produce ill-formed conformations, PAFlow is still able to generate molecules with stable and chemically plausible geometries at both the local and global levels. In future work, we plan to incorporate physical rules and energy-based constraints to further improve molecular structural quality.

Table 8: Results of molecular conformation evaluation. ($\downarrow$) denotes that lower values indicate better performance.

| Methods | Length ($\downarrow$) | Angle ($\downarrow$) | Torsion ($\downarrow$) |
|---|---|---|---|
| AR | 0.554 | 0.467 | 0.519 |
| FLAG | 0.511 | 0.284 | - |
| IPDiff | 0.402 | 0.415 | 0.386 |
| ALiDiff | 0.445 | 0.422 | 0.422 |
| PAFlow | 0.507 | 0.461 | 0.424 |

## F Discussion

### F.1 Difference Between PAFlow and Other FM-Based Methods

Flow Matching is a simulation-free approach for stably training continuous normalizing flows that demonstrates strong generative capabilities. Various types of probability paths can be used within the FM framework as summarized in Table 1 of [29], and the choice can be flexibly adapted based on the specific task requirements.

Table 9: Ratios of ring sizes for reference ligands and molecules generated by different methods.

| Ring Size | Ref | IPDiff | TAGMoL | PAFlow |
|---|---|---|---|---|
| 3 | 1.7% | 0.0% | 0.0% | 0.0% |
| 4 | 0.0% | 3.2% | 4.0% | 3.4% |
| 5 | 30.2% | 25.8% | 29.5% | 19.6% |
| 6 | 67.4% | 42.4% | 39.0% | 38.4% |
| 7 | 0.7% | 18.6% | 19.7% | 26.0% |
| 8 | 0.0% | 7.0% | 5.9% | 9.2% |
| 9 | 0.0% | 2.9% | 1.9% | 3.4% |

Table 10: Results on molecular conformation stability. For SE, the 25th percentile, median, and 75th percentile are reported.

| Methods | SE ($\downarrow$) | | | Clash ($\downarrow$) | |
|---|---|---|---|---|---|
| | 25% | 50% | 75% | Avg | Med |
| TargetDiff | 363 | 1233 | 13773 | 10.77 | 7.00 |
| ALiDiff | 1607 | 56055 | 5779799 | 8.68 | 5.00 |
| KGDiff | 566 | 84186 | 124750452 | 7.82 | 5.00 |
| PAFlow | 3339 | 90239 | 5906610 | 7.74 | 5.00 |

FlowSBDD[14] employs Rectified Flow to learn the transport mapping of atom coordinates and types from the prior to the data distribution, aiming for straight-line trajectories between the two. FlexSBDD[51] adopts a similar modeling approach. While such straight-line paths are computationally efficient, they lack the capacity to model complex generation tasks effectively. In contrast, PAFlow utilizes different probability paths tailored to the distinct characteristics of atom coordinates and types. Specifically, continuous coordinates are modeled using Gaussian distributions with variance-preserving (VP) trajectories derived from diffusion models, while discrete atom types are modeled with categorical distributions, for which we specifically construct a FM formulation and conditional vector field (detailed derivations in Appendix A.2). In addition, we prove that the resulting generative process is consistent with SE(3)-transformation invariance, offering a theoretical justification for our modeling strategies (see Appendix A.1).

Under the same experimental settings, a comparison between the results of these methods is presented. As shown in the Table 11 below, PAFlow significantly outperforms FlexSBDD and FlowSBDD on all affinity–related metrics while maintaining reasonable molecular properties. It is worth noting that FlexSBDD generates molecules with modeling protein flexibility, whereas PAFlow performs generation based on rigid proteins. Despite this difference, PAFlow still achieves superior performance, suggesting the greater effectiveness of our modeling strategy. Overall, although both PAFlow and FlexSBDD are built within the FM framework, they differ fundamentally in the design of probability paths and modeling strategies. We hope these extensions offer useful insights and contribute meaningfully to the advancement of FM-based molecular generation.

Table 11: Comparison of PAFlow with other FM-based methods. The table presents the mean values of each metric. The best results are highlighted in bold.

| Methods | Vina Score ($\downarrow$) | Vina Min ($\downarrow$) | Vina Dock ($\downarrow$) | High Affinity ($\uparrow$) | QED ($\uparrow$) | SA ($\uparrow$) | Div ($\uparrow$) |
|---|---|---|---|---|---|---|---|
| TargetDiff | -5.47 | -6.64 | -7.80 | 58.1% | 0.48 | 0.58 | 0.72 |
| FlowSBDD | -3.62 | -6.72 | -8.50 | 63.4% | 0.47 | 0.51 | 0.75 |
| FlexSBDD | -6.64 | -8.27 | -9.12 | 78.5% | **0.58** | **0.69** | **0.76** |
| **PAFlow** | **-8.31** | **-8.79** | **-9.46** | **80.8%** | 0.49 | 0.57 | 0.71 |

### F.2  Reasons for the Better Binding Affinity

There are two main reasons why the molecules generated by PAFlow achieve lower Vina scores compared to the ground truth in the test set. Firstly, we calculate the average and median Vina scores of the protein–ligand pairs in the training set to be $-8.19$ and $-8.32$ respectively, which are comparable to those of the molecules generated by PAFlow. This indicates that PAFlow effectively learned the binding patterns contained in the training data. Secondly, the interaction predictor is employed to guide the generation process. Since the predictor is trained using the Vina scores of the training set as labels, it captures prior domain knowledge about protein–ligand binding embedded in these scores. By incorporating this learned prior into the generation process, the model benefits not only from the information in the training data but also from the guidance, which together enable PAFlow to generate molecules with better binding affinity than those in the training set. Actually, the reference ground-truth ligands are excellent binders for their corresponding protein target according to certain experiments or predictions, but they may not be the optimal ones with the best Vina scores for the target.

### F.3  Limitations, Future Work and Broader Impact

While PAFlow demonstrates outstanding performance, there remain several potential limitations that we aim to address in future work. First, the volume and surface area of binding pockets used to train the atom number predictor are computed using PyKVFinder, which provides only an approximate estimation and may introduce bias. In the future, we plan to incorporate experimentally measured pocket information to improve the performance of the predictor. Additionally, we have attempted to apply flow matching to rigid proteins and observed promising results. We intend to extend this approach to flexible proteins, which better reflect realistic scenarios in structural biology. Lastly, since PAFlow only focuses on optimizing binding affinity, we plan to extend the guidance strategy to include molecular properties for multi-objective optimization to generate ligands with higher pharmaceutical potential.

Our work holds promise for improving the efficiency of drug design and advancing the pharmaceutical industry. It can help streamline the drug development process, reducing both time and resource costs. In addition, regulatory oversight of structure-based drug design (SBDD) technique is necessary to ensure they are used for socially beneficial purposes and to prevent potential misuse that could lead to the generation of harmful molecules.

### F.4  More Examples

The visualization of more ligand molecules generated by PAFlow, comparing to reference, ALiDiff and MolCRAFT, are provided in Fig. 11. We also present several unreasonable molecules generated by PAFlow as edge cases in Fig. 10 to illustrate its limitations, including improper double bonds, large rings, and fused rings, which occasionally occur. Incorporating physical rules and energy constraints is expected to alleviate this issue and improve the structural quality of the generated molecules.

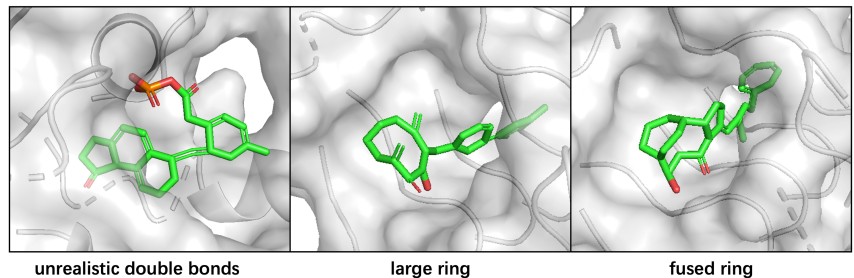

Figure 10: Some edge cases of molecules generated by PAFlow.

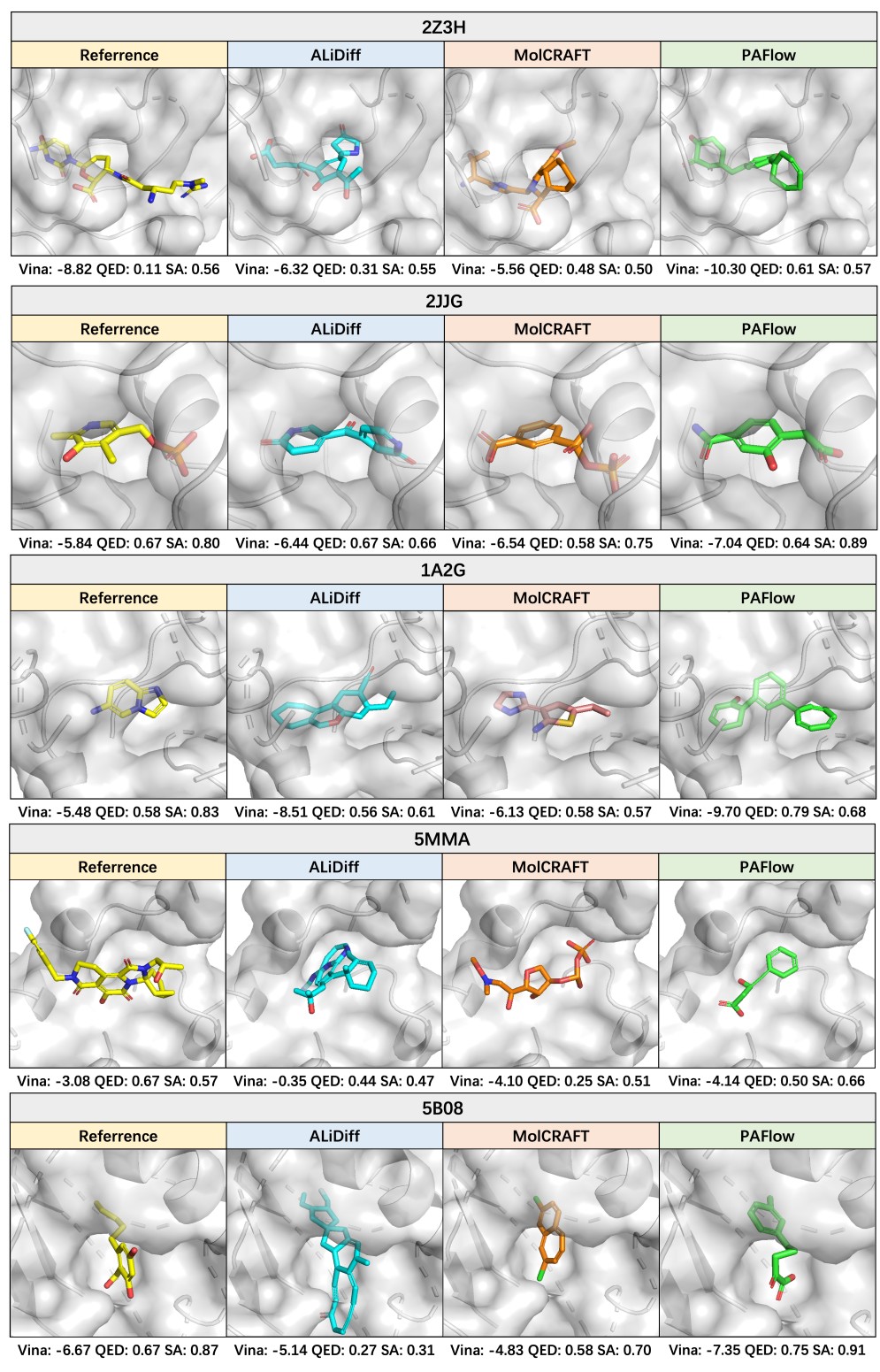

Figure 11: More visualizations of generated molecules and reference ligands for protein pockets. Carbon atoms of the reference ligands and molecules generated by AliDiff, MolCRAFT, and PAFlow are colored yellow, blue, orange, and green respectively. The corresponding Vina Score, QED, and SA for each molecule are also reported.

