# OpenReview forum: "Prior-Guided Flow Matching for Target-Aware Molecule Design with Learnable Atom Number"
_NeurIPS.cc/2025/Conference — NeurIPS 2025 poster_

### Official Review · Reviewer_CJX1 · 2025-06-28

**Clarity:** 2
**Significance:** 2
**Originality:** 2
**Rating:** 4
**Confidence:** 4

**Summary:**

This paper proposes PAFlow, a novel flow matching (FM)-based generative model for structure-based drug design (SBDD) that addresses key limitations of existing autoregressive and diffusion-based approaches, such as unnatural generation orders, unstable denoising trajectories, and reliance on predefined ligand priors for atom number prediction. PAFlow introduces a conditional flow matching (CFM) method for discrete atom types alongside variance-preserving (VP) paths for continuous coordinate generation, integrates a protein-ligand interaction predictor to enhance binding affinity, and employs a geometry-aware atom number predictor to eliminate dependency on reference ligands. Experiments on CrossDocked2020 demonstrate state-of-the-art performance, achieving an average Vina score of -8.31 (a 1.24 improvement over baselines) while maintaining favorable molecular properties, establishing FM as a robust alternative for 3D molecule generation in drug discovery.

**Questions:**

See in weakness.

**Ethical Concerns:**

["NO or VERY MINOR ethics concerns only"]

**Final Justification:**

Most of my concerns are addressed.

**Limitations:**

Yes

**Quality:**

2

**Strengths And Weaknesses:**

## Strengths
1. The proposed method demonstrates outstanding performance in terms of binding affinity scores on the CrossDocked2020 benchmark, surpassing existing baselines by a substantial margin.
2. The flow-based generative framework employed in the model exhibits superior sampling efficiency compared to diffusion-based counterparts, as evidenced by the reported generation time.
## Weaknesses:
1. Although the model achieves high predicted Vina scores under classifier guidance, it is well-known that such guidance often leads to ill-formed molecular conformations, such as abnormal bond lengths, distorted bond angles, and overly twisted structures. The paper lacks essential evaluation metrics that are standard in recent works [1,2,3], including bond length/angle JSD, validity, ring size distribution, strain energy, and atom clash statistics.
2. The authors claim that the learnable atom number predictor addresses the mismatch between ligand size and binding pocket geometry caused by prior-based methods. However, the experimental section does not include any results or ablation studies to validate this claim.
3. Prior research [4] has established a strong positive correlation between ligand size (i.e., number of atoms) and binding affinity. It remains unclear whether the performance gain attributed to the proposed atom number predictor stems merely from generating larger molecules. The paper does not analyze whether the predicted atom counts differ significantly from traditional methods or how this affects binding affinity.
4. The protein–ligand interaction predictor used for classifier guidance appears to be functionally equivalent to the affinity predictor introduced in TAGMol. Given this similarity, it is questionable to position this component as a major contribution of the paper.

References:

[1] Guan et al., DecompDiff: Diffusion Models with Decomposed Priors for Structure-Based Drug Design. arXiv:2403.07902

[2] Zheng et al., PoseCheck: Robust Pose Filtering for Diffusion-Based Molecular Docking. arXiv:2308.07413

[3] Qu et al., MolCraft: Structure-based Drug Design in Continuous Parameter Space. arXiv:2404.12141

[4] Wang et al., CBGBench: Fill in the Blank of Protein-Molecule Complex Binding Graph. arXiv:2406.10840

---

> ### Author Rebuttal · Authors · 2025-07-31
>
> **Q1**: Lack of structural evaluation metrics.
>
> **A1**: Thank you for your valuable comments. We report the average Jensen-Shannon divergence (JSD) between the bond length, bond angle, and torsion angle distributions of the generated and reference molecules, where a lower JSD indicates a distribution closer to that of real structures. In addition, we provide the ring size distribution and validity of generated molecules. Together, these evaluations comprehensively assess the substructure stability. To evaluate global structural stability, we also report strain energy and steric clash statistics, including the 25th, 50th (median), and 75th percentiles. As shown in the tables, PAFlow exhibits comparable performance to existing methods across both local and global structural metrics. Although classifier guidance is known to sometimes produce ill-formed conformations, PAFlow is still able to generate molecules with stable and chemically plausible geometries at both the local and global levels. We sincerely appreciate you pointing out this important issue and will include the above evaluation metrics in the revised manuscript. In future work, we plan to incorporate physical rules and energy-based constraints to further improve molecular structural quality.
>
> |Methods|Length ↓|Angle ↓|Torsion ↓|
> |-|-|-|-|
> |AR|0.554|0.467|0.519|
> |FLAG|0.511|0.284|-|
> |IPDiff|0.402|0.415|0.386|
> |ALiDiff|0.445|0.422|0.422|
> |PAFlow|0.507|0.461|0.424|
>
> |**Ring Size**|Ref|IPDiff|TAGMol|PAFlow|
> |-|-|-|-|-|
> |3|1.7%|0.0%|0.0%|0.0%|
> |4|0.0%|3.2|4.0|3.4|
> |5|30.2%|25.8|29.5|19.6|
> |6|67.4%|42.4|39.0|38.4|
> |7|0.7%|18.6|19.7|26.0|
> |8|0.0%|7.0|5.9|9.2|
> |9|0.0%|2.9|1.9|3.4|
> |**Validity**|-|98.3%|92.6%|94.7%|
>
> |Methods||SE ↓| |Clash ↓| |
> |-|-|-|-|-|-|
> | |25%|50%|75%|Avg|Med|
> |TargetDiff|363|1233|13773|10.77|7.00|
> |ALiDiff|1607|56055|5779799|8.68|5.00|
> |KGDiff|566|84186|124750452|7.82|5.00|
> |PAFlow|3339|90239|5906610|7.74|5.00|
>
> **Q2**: Lack of validation for atom number predictor.
>
> **A2**: Thank you for your question. In Tab. 4 on page 9, we compare the results of two approaches: sampling atom numbers from the predefined distribution (i.e., the prior-based method referred to as "predefined" in the table) and using the atom number predictor (the last two rows in the table). As shown in the results, molecules generated using the predictor achieve substantial improvements over the prior-based method in both affinity-related metrics and molecular properties, while maintaining diversity. To more intuitively demonstrate the effectiveness of our predictor, we compare it with the predefined sampling by generating 1,000 atom numbers for each of three randomly selected test proteins. For each protein, we define ±20% and ±30% intervals around the reference atom number and compute the proportion of generated samples falling within these ranges. As shown in table, a considerable portion of the results from the predefined sampling fall outside the ±30% range, which may lead to overly large or small molecules that poorly match the pocket geometry. In contrast, the samples acquired by our predictor are concentrated within the ±20% range, suggesting a stronger potential for generating molecules that are structurally compatible with the target pocket.
>
> | | |<70%|[70%, 80%]|[80%, 100%]|[100%, 120%]|[120%, 130%]|>130%|
> |-|-|-|-|-|-|-|-|
> |4YHJ|Predefined|4.7%|10.9%|26.2%|34.2%|12.5%|11.5%|
> | |Predict|0|0|64.0%|36.0%|0|0|
> |2E24|Predefined|7.4%|7.8%|17.1%|17.4%|13.7%|36.6%|
> | |Predict|0|0|2.9%|81.4%|15.5%|0.2%|
> |3NFB|Predefined|25.2%|11.1%|29.8%|23.1%|5.8%|5.0%|
> | |Predict|0|0.4%|97.2%|2.4%|0|0|
>
> **Q3**: It remains unclear whether the performance gain attributed to the proposed atom number predictor stems merely from generating larger molecules. The paper does not analyze whether the predicted atom counts differ significantly from traditional methods or how this affects binding affinity.
>
> **A3**: Thank you for your valuable suggestion. Although larger molecules tend to exhibit higher binding affinity, the performance gain in our method does not stem from simply generating molecules with more atoms. Instead, it results from generating molecules whose atom counts are closer to the reference ligand. As shown in the table in **A2**, most atom numbers predicted by our model fall within the ±20% range of the reference ligand’s atom count, without producing significantly larger molecules. In contrast, the traditional method—which samples atom numbers from predefined distributions—yields a substantial portion of results outside the ±30% range. Such deviation increases the likelihood of generating molecules that are either too large or too small to fit well within the binding pocket, thereby negatively affecting the protein–ligand interaction.
>
> The traditional method employs a set of predefined atom number distributions, each associated with a specific interval of pocket space size. When determining the number of atoms to generate, the space size of the given pocket is first calculated, and then a corresponding distribution is selected based on which interval the space size falls into. This coarse-grained approach makes it difficult to accurately capture the appropriate atom number range for molecules targeting different pockets. To address this limitation, our proposed predictor estimates the atom number based on multiple geometric features of the pocket, including the pocket atom number, volume, surface area, and space size. As shown in Fig. 9 (page 19), these factors are strongly correlated with the atom number of the reference ligands. By leveraging this pocket-specific information, our predictor provides a smoother and more informed estimation strategy, which has a higher potential to yield molecules that effectively interact with the target proteins. We will include the relevant analysis in the revised version of the paper.
>
> **Q4**: The protein–ligand interaction predictor used for classifier guidance appears to be functionally equivalent to the affinity predictor introduced in TAGMol. Given this similarity, it is questionable to position this component as a major contribution of the paper.
>
> **A4**: Thank you for your comment. We acknowledge that both the interaction predictor used in PAFlow and the affinity predictor in TAGMol can be categorized as classifier guidance. However, their underlying principles and effects differ significantly. TAGMol is built upon a diffusion-based framework, where the probability path of atom coordinates is defined as $q(x_t|x_0)=\mathcal{N}(\sqrt{\bar{\alpha_t}}x_0,(1-\bar{\alpha_t})\mathbf{I})$, with $t=0$ corresponding to data and $t=T$ to the prior. The generative process without guidance in TAGMol is defined as $q(x_{t-1}|x_t,x_0)=\mathcal{N}(\tilde{\mu_t},\tilde{\beta_t} \mathbf{I})$, where $\tilde{\mu_t}=\frac{\sqrt{\bar{\alpha_{t-1}}}\beta_t}{1-\bar{\alpha_t}} x_0+\frac{\sqrt{\alpha_t}(1-\bar{\alpha_{t-1}})}{1-\bar{\alpha_t}}x_t$, $\tilde{\beta_t}=\frac{1-\bar{\alpha_{t-1}}}{1-\bar{\alpha_{t}}}\beta_t$. In generation, the coordinate difference between two consecutive timesteps is $\mathrm{d}x=x_{t-1}-x_t=\tilde{\mu_t}-\tilde{\mu_{t+1}} + \sqrt{{\tilde{\beta_t}}+ {\tilde{\beta_{t+1}}}} \mathbf{I}$. By substituting the expression of $\tilde{\mu}$, we can simplify the result to
>
> $\mathrm{d}x=(\sqrt{\bar{\alpha_{t-1}}}-\sqrt{\bar{\alpha_t}})x_0+\sqrt{\sigma_{t-1}^2+\sigma_t^2+{\tilde{\beta_t}}+{\tilde{\beta_{t+1}}}}\mathbf{I}$,
>
> where $\sigma_{t}=\frac{\sqrt{\alpha_{t+1}}(1-\bar{\alpha_t})}{\sqrt{1-\bar{\alpha_{t+1}}}}$. In PAFlow, the generative process is defined with $t=1$ corresponding to data and $t=0$ to the prior. We discretize and reverse the time steps to align with the time series of TAGMol to facilitate the comparison. According to Eq.5 on page 4, we obtain: $\mathrm{d}x =\frac{\sqrt{\bar{\alpha_{t}}}-\sqrt{\bar{\alpha_{t-1}}}}{1-\bar{\alpha_t}}(\sqrt{\bar{\alpha_t}}x_t-x_0)$. By substituting $x_t=\sqrt{\bar{\alpha_t}}x_0 + \sqrt{1-\bar{\alpha_t}}$, the expression can be simplified as:
>
> $\mathrm{d}x =(\sqrt{\bar{\alpha_{t}}}-\sqrt{\bar{\alpha_{t-1}}})x_0+\frac{\bar{\alpha_{t-1}}(\alpha_t-\sqrt{\alpha_t})}{\sqrt{1-\bar{\alpha_t}}}\mathbf{I}$.
>
> Clearly, without guidance, both methods share the same mean in their $\mathrm{d}x$ expressions. By plotting the variance curves, we observe that PAFlow exhibits a much smaller variance term than TAGMol, resulting in a smoother and more stable generation trajectory.
>
> When guidance is applied, the generation in TAGMol is defined as $q(x_{t-1}|x_t,x_0)=\mathcal{N}(\tilde{\mu_t}+\tilde{\beta_t}\nabla \log p_{\theta}(y|x_t), \tilde{\beta_t} \mathbf{I})$, and $\mathrm{d}x$ can be expressed as:
>
> $\mathrm{d}x=(\sqrt{\bar{\alpha_{t-1}}}-\sqrt{\bar{\alpha_t}})x_0 + \frac{1-\bar{\alpha_{t-1}}}{1-\bar{\alpha_{t}}}\beta_t\nabla\log p_{\theta}(y|x_t)+\sqrt{\sigma_{t-1}^2+\sigma_t^2+{\tilde{\beta_t}}+ {\tilde{\beta_{t+1}}}}\mathbf{I}$.
>
> For PAFlow with guidance, based on Eq.13 on page 5, $\mathrm{d}x$ can be written as:
>
> $\mathrm{d}x =(\sqrt{\bar{\alpha_t}}-\sqrt{\bar{\alpha_{t-1}}})x_0+\frac{\sqrt{\bar{\alpha_t}}-\sqrt{\bar{\alpha_{t-1}}}}{2\bar{\alpha_t}}\nabla\log p_{\phi}(y|x_t)+\frac{\bar{\alpha_{t-1}}(\alpha_t-\sqrt{\alpha_t})}{\sqrt{1-\bar{\alpha_t}}}\mathbf{I}$.
>
> It is evident that after incorporating guidance, the mean paths of PAFlow and TAGMol differ, indicating that the two methods employ mathematically distinct guidance strategies. Furthermore, as shown in Tab. 1 on page 7, molecules generated by PAFlow significantly outperform those of TAGMol across all affinity-related metrics, demonstrating the greater effectiveness of PAFlow’s guidance approach. The smoother probability path of PAFlow without guidance also contributes to improved stability during guidance.
>
> Overall, although both PAFlow and TAGMol utilize classifier guidance, they differ fundamentally in their mathematical formulation and performance.

---

> > ### Comment · Reviewer_CJX1 · 2025-08-08
> >
> > Thank you for your responses, which address most of my concerns. I will raise my score, good luck!

---

> > > ### Author Response · Authors · 2025-08-08
> > >
> > > We sincerely appreciate your response and recognition of our work! Please do not hesitate to contact us if you have any further questions.

---

### Official Review · Reviewer_cwgo · 2025-07-01

**Clarity:** 3
**Significance:** 2
**Originality:** 2
**Rating:** 5
**Confidence:** 3

**Summary:**

The authors aim to solve the problem of structure-based drug design. Upon this, the authors noticed limitations of existing generative methods, such as unstable probability dynamics and molecular size mismatches. To address such limitations, the authors introduce PAFlow, a novel target-aware molecular generation model designed for structure-based drug design. PAFlow addresses such limitations by leveraging the flow matching framework with a new conditional flow matching for discrete atom types, incorporating a protein-ligand interaction predictor for guiding generation towards high-affinity regions. Additionally, it integrates a learnable atom number predictor, relying solely on protein pocket information, which significantly improves the alignment of generated molecule size with target geometry, achieving state-of-the-art binding affinity results while maintaining favorable molecular properties.

**Questions:**

- Looking at Figure 5, the generation trajectory of the target diff appears much messier compared to the proposed method, yet the sampling efficiency shows slightly better results than the proposed method. How can we understand this phenomenon?
- What insights might there be if coordinate and atom type probability density paths were modeled jointly rather than separately?
- Regarding prior guidance, how dependent is the method on prior quality?

**Ethical Concerns:**

["NO or VERY MINOR ethics concerns only"]

**Final Justification:**

The paper proposes a solid framework to solve the SBDD problem with extensive evaluation to prove superiority over prior works, and the authors' rebuttal well addressed my concerns.

**Limitations:**

Yes

**Paper Formatting Concerns:**

I have found no formatting issues.

**Quality:**

3

**Strengths And Weaknesses:**

## Strengths

- Extensive evaluation and good performance: The paper contains a comprehensive comparison with a number of baselines on a large-scale benchmark with comprehensive metrics, and necessary ablation studies of the proposed method, which helps to understand the empirical robustness of the proposed method.
- Overall decent recipe to solve SBDD: The proposed method appears to be a well-developed methodology that efficiently utilizes prior work formulations (probability density path formulation, prior guidance) to solve current SBDD problems, while adding unique methods (proposition, learnable atom predictor) to overcome the limitations of prior work.
- Efficient improvement over previous methods: The introduction of an atom predictor effectively resolves the issue that non-autoregressive previous works for SBDD problems required reference information for generated ligands. In addition, the sampling efficiency over baseline is comparable to the most efficient baselines.

## Weaknesses

- Incremental methodological contribution: The method appears to be largely similar in framework to [1] in terms of using flow matching and prior guidance, and to [2] in terms of probability density path formulation, suggesting that the novelty may be limited to incremental improvements.
- Lack of formulation justification: In Equation (4), the probability density is defined as the product of probability distributions of atom type and atom coordinate. Given that these two features have strong correlations with each other, it would seem more natural to formulate it as $p_t(x, a| x_1, a_1, p)$. Is there a reason why this approach is difficult? The text simply explains that it follows [3] without specific justification, which seems insufficient beyond merely following prior work.
- Typo:
    - In line 143, it appears that the definition of $\hat{\alpha}$ is missing.

[1] Z. Zhang, M. Zitnik, and Q. Liu, “Generalized protein pocket generation with prior-informed flow matching,” arXiv preprint arXiv:2409.19520, 2024.

[2] J. Guan, W. W. Qian, X. Peng, Y. Su, J. Peng, and J. Ma, “3d equivariant diffusion for target349 aware molecule generation and affinity prediction,” arXiv preprint arXiv:2303.03543, 2023.

[3] Y. Song, J. Gong, M. Xu, Z. Cao, Y. Lan, S. Ermon, H. Zhou, and W.-Y. Ma, “Equivariant flow
407 matching with hybrid probability transport for 3d molecule generation,” Advances in Neural
 Information Processing Systems, vol. 36, pp. 549–568, 2023.

---

> ### Author Rebuttal · Authors · 2025-07-31
>
> **Q1**: Incremental methodological contribution
>
> **A1**: Thank you for your comment. While both PAFlow and PocketFlow [1] are built upon the flow matching (FM) framework, their implementation details are entirely different. FM is a simulation-free approach for stably training continuous normalizing flows (CNFs), and various types of probability paths can be employed within this framework, as summarized in Tab. 1 of [3]. PocketFlow is designed for generating protein pockets conditioned on given ligand molecules, which can be considered a dual problem of structure-based drug design (SBDD). It models the optimal transport mapping from the prior to the data distribution, aiming for straight-line trajectories between them. Similar modeling strategies have also been adopted in SBDD tasks by methods such as FlexSBDD and FlowSBDD. Although these straight-line paths offer shorter transport distances and computational efficiency, they lack the capacity to effectively model complex tasks such as SBDD. In contrast, PAFlow uses different probability trajectories tailored to the distinct characteristics of atom coordinates and types. For continuous atomic coordinates, we employ the variance-preserving (VP) path derived from diffusion models. For discrete atom types, we model them using categorical distributions and specifically construct a corresponding FM formulation and conditional vector field (see lines 144–149 on page 4, detailed derivations in Appendix A.2). Moreover, we demonstrate that the resulting generative process is SE(3)-transformation invariant (Appendix A.3), providing a theoretical justification for the soundness of our modeling approach. We compare these methods under the same experimental settings. As shown in the table, PAFlow significantly outperforms both FlowSBDD and FlexSBDD across all affinity-related metrics while maintaining favorable molecular properties, demonstrating the effectiveness of our modeling strategy.
>
> |Methods|Vina Score↓|Vina Min ↓|Vina Dock ↓|High Affinity ↑|QED ↑|SA ↑|Div ↑|
> |-|-|-|-|-|-|-|-|
> |TargetDiff|-5.47|-6.64|-7.80|58.1%|0.48| 0.58|0.72|
> |FlowSBDD|-3.62|-6.72|-8.50|63.4%|0.47|0.51|0.75
> |FlexSBDD|-6.64|-8.27|-9.12|78.5%|0.58|0.69|0.76
> |PAFlow|**-8.31**|**-8.79**|**-9.46**|**80.8%**|0.49|0.57|0.71
>
> PocketFlow also incorporates prior guidance during generation, and both PocketFlow and PAFlow use predictors to guide the process. However, the underlying probability trajectories are fundamentally different, leading to distinct mathematical formulations and derivations. In the derivation, PocketFlow starts from a general ODE formulation, while PAFlow is based on the guided vector field for Gaussian trajectory. Regarding implementation, the predictor in PocketFlow is trained on binary labels (0 for below-average affinity and 1 for above-average affinity) aiming to estimate the probability that affinity equals 1. In contrast, PAFlow uses vina scores normalized to the [0, 1] range as labels and directly predicts affinity values. This allows PAFlow to learn a smoother affinity distribution, enabling more accurate and fine-grained guidance during the generation process.
>
> In terms of the probability density path, PAFlow modeling under the FM framework adopts the Gaussian trajectory used in [2], where TargetDiff is introduced as a diffusion-based method. We can mathematically prove that when using the same Gaussian path, FM and diffusion share the same mean trajectory, but FM exhibits significantly lower variance (Please let us know if you need the full proof). As a result, the denoising trajectories in diffusion (Fig. 5a) tend to be highly stochastic, leading to unstable probability dynamics and high computational cost. In contrast, the FM-based path (Fig. 5b) is much smoother and more stable due to its low variance. This smoother path enables PAFlow to reduce the number of sampling steps and achieve a 5.5× speedup in efficiency (Page 8, Line 275; Fig. 4), while also delivering better generation quality (Tab. 2).
>
> Overall, although both PAFlow and [1] adopt the FM framework with prior guidance, our implementation differs substantially from [1]. While we share the same probability path as [2], the significantly lower variance of FM enables PAFlow to achieve a smoother and more stable generation process. We believe the improvements introduced in PAFlow are meaningful and technically valuable extensions for the SBDD task.
>
> **Q2**: Lack of formulation justification
>
> **A2**: Thank you for this valuable suggestion. We fully agree with the perspective that atom coordinates and atom types are inherently interdependent in molecular structures, and that jointly modeling them is more reasonable. However, directly modeling the joint distribution $p_t(x, a \mid x_1, a_1, p)$ faces several practical challenges. Specifically, atom coordinates are 3D continuous variables, while the atom types of ligands and proteins are 27-dimensional and 13-dimensional discrete variables, respectively. Learning a joint distribution in such a high-dimensional space with heterogeneous dimensions and different modalities introduces computational complexity and lacks effective modeling mechanisms. Existing flow matching methods also lack theoretical foundations and practical implementations for constructing mixed-modality vector fields. Therefore, to facilitate both the modeling and the computation of the vector field, we factorize the joint probability into the product of a Gaussian distribution and a categorical distribution, allowing us to separately handle the 3D continuous coordinates and high-dimensional discrete atom types.
>
> It is important to emphasize that we do not assume independence between atom coordinates and types. Instead, they are conditioned on the same contextual information and their interdependence is considered in the network architecture. We employ an SE(3)-Equivariant GNN for modeling, where atom types are associated with nodes and coordinates with edges. Through the message passing mechanism in the graph structure, the two components influence each other. During the generation process, the spatial positions of nodes and the types of edges are updated jointly, which implicitly captures the dependencies between atom coordinates and atom types. This modeling strategy has been adopted by most existing non-autoregressive structure-based drug design methods, and our experiments also demonstrate its effectiveness in achieving strong performance. Joint modeling undoubtedly presents a valuable research direction that could significantly advance the field of drug design, and we are willing to further explore it in future work.
> **Q3**: Looking at Figure 5, the generation trajectory of the targetdiff appears much messier compared to the proposed method, yet the sampling efficiency shows slightly better results than the proposed method. How can we understand this phenomenon?
>
> **A3**: As shown in Fig. 5(a), the unstable probability dynamics of diffusion models lead to noisy and disordered generation trajectories. In contrast, Fig. 5(b) demonstrates that the flow matching approach produces much smoother trajectories by updating molecular positions through the vector field. Such smooth trajectories enable PAFlow to reduce the number of sampling steps while preserving generation quality. As illustrated in Fig. 4, PAFlow requires only 717s on average to generate 100 molecules, whereas TargetDiff takes 4009s, yielding a 5.59× speedup. Regarding the quality of generated molecules, Tab. 1 shows that PAFlow substantially outperforms TargetDiff on all binding-related metrics, while maintaining comparable molecular properties. These results clearly demonstrate that PAFlow achieves both efficient sampling and high-quality generation.
>
> **Q4**: What insights might there be if coordinate and atom type probability density paths were modeled jointly rather than separately?
>
> **A4**: Thank you for the thought-provoking question. Jointly modeling the coordinate and atom type probability densities could indeed provide a more expressive framework that captures deeper interactions between molecular geometry and atom semantics. It has great potential to lead to enhanced chemical validity and structural coherence in the generated molecules. As discussed in **A2**, joint modeling faces many computational challenges due to the different modalities and heterogeneous dimensions of the data. Nevertheless, this remains a valuable direction for future research, and we appreciate your insightful comment highlighting its importance.
>
> **Q5**: Regarding prior guidance, how dependent is the method on prior quality?
>
> **A5**: Thank you for your question. According to [5], the CrossDocked2020 dataset mainly consists of complexes with moderate binding affinities, so the prior quality we use is not particularly high. As shown in Tab. 3, the generated molecules exhibit substantial improvements across all affinity-related metrics when prior guidance is employed, compared to the unguided setting. This demonstrates that even our predictor trained on moderate-quality data can effectively guide the generation process, indicating a certain level of robustness to the prior. In addition, high-quality prior information can help the predictor capture more accurate protein–ligand binding patterns, thereby providing more precise guidance during generation. Designing more effective priors through better data or modeling is an important direction for further improvement.
>
> [1] Generalized protein pocket generation with prior-informed flow matching.
>
> [2] 3d equivariant diffusion for target-aware molecule generation and affinity prediction.
>
> [3] Improving and generalizing flow-based generative models with minibatch optimal transport.
>
> [4] Equivariant flow matching with hybrid probability transport for 3d molecule generation.
>
> [5] Tagmol: Target-aware gradient-guided molecule generation.

---

> > ### Comment · Reviewer_cwgo · 2025-08-04
> >
> > Thank you for the author's detailed response.
> > The author has effectively addressed my concerns and questions.
> > While I acknowledge their rebuttal regarding my concern about incremental novelty, at the property level, I still find that in the current version, the novelty does not warrant a higher score. Therefore, I will maintain my original assessment.

---

> > > ### Author Response · Authors · 2025-08-06
> > >
> > > Thank you very much for your reply and recognition of our responses. We appreciate your valuable suggestions and will incorporate the discussions and results to the final version. Please let me know if you have any other questions.

---

### Official Review · Reviewer_ns3U · 2025-07-02

**Clarity:** 2
**Significance:** 2
**Originality:** 2
**Rating:** 3
**Confidence:** 5

**Summary:**

This paper introduces PAFlow, a novel target-aware generative model for structure-based drug design. PAFlow integrates prior protein-ligand interaction knowledge and a learnable atom number predictor into a conditional flow matching framework to generate 3D molecules with improved binding affinity. The model addresses ligand-pocket size mismatches and guides generation toward higher-affinity regions.

**Questions:**

- How did you obtain the binding affinity labels to train the guidance network? How do you ensure the affinities are comparable across different targets?
- How does the model perform on molecular stability metrics, such as bond length accuracy?
- How is pocket size calculated in Appendix E.5?

**Ethical Concerns:**

["NO or VERY MINOR ethics concerns only"]

**Final Justification:**

Authors have addressed my concerns and questions regarding the validity of the generated molecules and provided additional experiments. I really appreciated their efforts.
However, regarding the novelty of using flow matching frameworks and training an additional Vina score predictor for guided generation, the originality of this work is still limited. As a result, I will maintain my rating.

**Limitations:**

yes

**Quality:**

2

**Strengths And Weaknesses:**

**Strength**
This paper addresses the problem of undetermined atom numbers by introducing an additional predictor.
Instead of relying on explicit priors as in previous works, it trains a binding affinity predictor and uses it to guide flow matching sampling toward higher-affinity distributions.
**Weakness**
The novelty of the flow matching framework is limited, as flow matching has already been applied to SBDD in prior works (e.g., *FlexSBDD: Structure-Based Drug Design with Flexible Protein Modeling*, NeurIPS 2024).
Moreover, although the authors claim their method generates stable molecules of high quality, the visualized results in Figure 3 include unrealistic double bonds. The work should include more results on bond length and bond angle accuracy as evaluation metrics.
While the method generates molecules with low Vina scores, these scores are even lower than the ground truth from the test dataset, which requires discussion in the paper.

---

> ### Author Rebuttal · Authors · 2025-07-31
>
> **Q1**: The novelty of the flow matching framework is limited, as flow matching has already been applied to SBDD in prior works (e.g., FlexSBDD).
>
> **A1**: Thank you for raising this important point. Flow Matching (FM) is a simulation-free approach for stably training Continuous Normalizing Flows that exhibits strong generative performance. Within the FM framework, various probability paths can be used for modeling. Tab. 1 in [1] summarizes available options, including diffusion and optimal transport paths, with the appropriate choice depending on the specific application.
>
> Although both FlexSBDD and PAFlow are built upon the FM framework, their modeling strategies are fundamentally different. FlexSBDD employs the optimal transport displacement interpolant as the probability path for transferring atom coordinates and types, aiming for straight flows from prior to data. FlowSBDD[2] adopts the same modeling approach. While such paths offer shorter distances and reduced computational cost, they are limited in their capacity to effectively model complex tasks. In contrast, PAFlow utilizes different probability paths tailored to the different characteristics of atom coordinates and types. For continuous coordinates, we use the Variance Preserving (VP) path derived from diffusion models. For discrete atom types, we model them using categorical distributions and specifically design a novel FM formulation along with a conditional vector field (see lines 144–149 on page 4), where detailed derivations are provided in Appendix A.2. Furthermore, we demonstrate that this generation process is consistent with SE(3)-transformation invariance (see Appendix A.1), providing a theoretical justification for the validity of our modeling approach.
>
> Under the same experimental settings, a comparison between the results of these methods is presented. As shown in the table below, PAFlow significantly outperforms FlexSBDD on all affinity–related metrics while maintaining reasonable molecular properties. It is worth noting that FlexSBDD generates molecules with modeling protein flexibility, whereas PAFlow performs generation based on rigid proteins. Despite this difference, PAFlow still achieves superior performance, suggesting the greater effectiveness of our modeling strategy. Overall, although both PAFlow and FlexSBDD are built within the FM framework, their implementation details differ substantially. We believe that PAFlow introduces meaningful and technically valuable contributions in the design of the probability paths.
>
> | Methods | Vina Score↓ | Vina Min ↓ | Vina Dock ↓ | High Affinity ↑ | QED ↑ | SA ↑ | Div ↑ |
> |-|-|-|-|-|-|-|-|
> | TargetDiff | -5.47| -6.64| -7.80 | 58.1% | 0.48 | 0.58  | 0.72  |
> | FlowSBDD| -3.62| -6.72| -8.50| 63.4%  | 0.47 | 0.51 | 0.75
> | FlexSBDD| -6.64| -8.27 | -9.12 | 78.5% | 0.58| 0.69 | 0.76
> | PAFlow | **-8.31**  | **-8.79** | **-9.46** | **80.8%**| 0.49 | 0.57| 0.71|
>
> **Q2+Q6**: **Q2** Moreover, although the authors claim their method generates stable molecules of high quality, the visualized results in Figure 3 include unrealistic double bonds. The work should include more results on bond length and bond angle accuracy as evaluation metrics. **Q6** How does the model perform on molecular stability metrics, such as bond length accuracy?
>
> **A2+A6**:
> Thank you for your valuable comment. We report the average Jensen-Shannon divergence (JSD) between the bond length, bond angle, and torsion angle distributions of the generated and reference molecules to assess conformational stability. The JSD is computed between the estimated distributions from generated and reference molecules, where a lower JSD indicates closer alignment with realistic molecular geometries. As shown in the table, PAFlow achieves stable performance across all metrics, indicating the geometric reliability of the molecular structures.
>
> We acknowledge that a few visualized molecules may exhibit unrealistic double bonds, which is likely due to the fact that we do not explicitly model chemical bonds but instead rely on OpenBabel for bond assignment. However, the overall statistical results suggest that the structural quality remains stable. We appreciate you pointing out this potential issue, and in future work, we plan to enhance the molecular structure modeling by incorporating bond representations and introducing energy-based constraints to improve chemical plausibility.
>
> |Methods|Length|Angle|Torsion|
> |-|-|-|-|
> |AR|0.554|0.467|0.519|
> |FLAG|0.511|0.284|-|
> |IPDiff|0.402|0.415|0.386|
> |ALiDiff|0.445|0.422|0.422|
> |PAFlow|0.507|0.461|0.424|
>
> **Q3**: While the method generates molecules with low Vina scores, these scores are even lower than the ground truth from the test dataset, which requires discussion in the paper.
>
> **A3**: Thank you for your valuable suggestion. There are two main reasons why the molecules generated by PAFlow achieve lower Vina scores compared to the ground truth in the test set. Firstly, we calculate the average and median Vina scores of the protein–ligand pairs in the training set to be -8.19 and -8.32 respectively, which are comparable to those of the molecules generated by PAFlow. This indicates that PAFlow effectively learned the binding patterns contained in the training data. Secondly, the interaction predictor is employed to guide the generation process. Since the predictor is trained using the Vina scores of the training set as labels, it captures prior domain knowledge about protein–ligand binding embedded in these scores. By incorporating this learned prior into the generation process, the model benefits not only from the information in the training data but also from the guidance, which together enable PAFlow to generate molecules with better binding affinity than those in the training set. Moreover, many existing methods [3-5] have achieved Vina scores better than the ground truth in the test set, which is a reasonable outcome resulting from effectively leveraging information of the training data. Actually, the reference ground-truth ligands are excellent binders for their corresponding protein target according to certain experiments or predictions, but they may not be the optimal ones with the best Vina scores for the target. We will include a corresponding discussion in the revised manuscript.
>
> **Q4**: How did you obtain the binding affinity labels to train the guidance network?
>
> **A4**: Thank you for your question. The binding affinity labels used to train the interaction predictor are Vina Scores computed by AutoDock Vina, following the setup in [4]. To facilitate stable learning, these scores are further normalized to the [0, 1] range using Min-Max normalization, where higher values indicate stronger binding affinity.
>
> **Q5**: How do you ensure the affinities are comparable across different targets?
>
> **A5**: Thank you for raising this point. We address this question in two parts.
>
> (1) Regarding the training process, the binding affinities used to train the property predictor are all obtained from Vina scores computed by AutoDock Vina. Since these scores are derived based on a consistent set of physicochemical principles, they share a unified numerical scale and physical interpretation across all protein–ligand pairs.
> Furthermore, we analyze the distribution of Vina scores in the training set below and observe substantial overlap across different protein–ligand pairs, with most scores falling within the range of [-12, -4]. This indicates that the binding affinities across different targets are defined on a consistent scale. In addition, prior works [4-6] have also adopted similar strategies of training on binding affinities across different targets.
>
> |Range|< -12|[-12,-4]|> -4|
> |-|-|-|-|
> |Percentage|4.40%|91.78%|3.82%|
>
> (2) For generation, the High Affinity in Tab. 1 measures the percentage of generated molecules that bind more strongly to a given target than the reference ligand, with molecules grouped according to their respective target proteins. As shown in Tab. 1, PAFlow significantly outperforms other methods in High Affinity, indicating its strong capability to consistently generate tightly binding molecules across different targets. Fig. 2 presents the median Vina binding energy of molecules generated by different methods for each of the 100 test proteins. PAFlow achieves the best results on 77% of the targets, which further suggests that PAFlow has the potential to maintain strong performance for a wide range of target proteins. Additionally, except for High Affinity, the other affinity-related metrics reported in Tab. 1 are calculated over molecules generated for all test proteins collectively, without restricting to individual targets. This evaluation approach is commonly adopted by most structure-based drug design methods [2-6], and therefore, we follow the same convention.
>
> **Q7**: How is pocket size calculated in Appendix E.5?
>
> **A7**: Thank you for your question. Pocket size is characterized using four features: the number of pocket atoms $N_P$, binding site volume $V$, binding site surface area $A$, and space size $S$. Among them, $N_P$ is obtained by directly counting the atoms in the pocket, while $V$ and $A$ are computed using pyKVFinder. The space size $S$ is the median value of the top 10 largest pairwise distances between protein atoms. These features can provide a comprehensive characterization of pocket size.
>
> [1] Improving and generalizing flow-based generative models with minibatch optimal transport.
>
> [2] Rectified Flow For Structure Based Drug Design.
>
> [3] Flexsbdd: Structure-based drug design with flexible protein modeling.
>
> [4] KGDiff: towards explainable target-aware molecule generation with knowledge guidance.
>
> [5] Tagmol: Target-aware gradient-guided molecule generation.
>
> [6] Multi-Objective Structure-Based Drug Design Using Causal Discovery.

---

> > ### Comment · Reviewer_ns3U · 2025-08-05
> >
> > Thank you for your thorough responses, which effectively address my concerns. I particularly appreciate how you incorporate an atom number predictor based on pocket information for molecule generation, as many approaches merely rely on the atom numbers of references. I do have one additional question: the validity of the bond angles and torsions could be improved. Do you have any suggestions or potential solutions to enhance these metrics? Is it reasonable to say that the guidance term is compromising valid chemical bond geometries by dragging the flow away from training distributions?

---

> > > ### Author Response · Authors · 2025-08-06
> > >
> > > Thank you for your continued engagement and thoughtful follow-up questions. We appreciate your recognition of the atom number predictor. While our method exhibits comparable structural stability to existing approaches, we fully agree that improving geometric validity is an important direction for further enhancement. Classifier guidance directs the generation toward regions of high binding affinity, but without explicitly modeling chemical bonds or considering structural stability, the resulting molecules may exhibit suboptimal geometric validity. This phenomenon has also been observed in other guided generation frameworks.
> > >
> > > To address this issue, we plan to explicitly model bond distributions in the same way as atom coordinates and types, enabling the model to better learn physically consistent patterns. Additionally, we will explore incorporating physical rules or energy-based constraints to further optimize the geometry during generation.
> > >
> > > Thank you again for highlighting this valuable point. We hope these clarifications and future directions help to further support the contributions of our work.

---

> > > ### Author Response · Authors · 2025-08-08
> > >
> > > Thanks for your time spent reviewing our work and the constructive suggestions. As the discussion period draws to a close, we would be grateful for your further feedback on our clarifications and let us know if you have any other concerns. We are more than willing to address any remaining questions.
> > >
> > > We would greatly appreciate it if you would consider adjusting the score, on the basis of our responses and other review comments.
> > >
> > > Thanks again for your valuable and thoughtful reviews!

---

### Official Review · Reviewer_AuA9 · 2025-07-03

**Clarity:** 3
**Significance:** 3
**Originality:** 3
**Rating:** 5
**Confidence:** 4

**Summary:**

This paper proposes PAFlow, a novel structure-based drug design (SBDD) model built on the flow matching (FM) framework. The method separately models continuous 3D atomic coordinates and discrete atom types using tailored probability paths and incorporates two key components: a protein-ligand interaction predictor that guides generation toward high-affinity regions, and a learnable atom number predictor that aligns molecule size to the target binding pocket. The method achieves strong results on CrossDocked2020, outperforming many baselines.

**Questions:**

All added within "Strengths And Weaknesses"

**Ethical Concerns:**

["NO or VERY MINOR ethics concerns only"]

**Final Justification:**

[Final Thoughts]: I have raised my score based on the sufficient author responses.

**Limitations:**

All added within "Strengths And Weaknesses"

**Quality:**

3

**Strengths And Weaknesses:**

The idea of using interaction prediction to steer the vector field during flow-based generation is clever and effective. It directly targets the main objective of SBDD, tight binding, instead of relying only on general chemical priors.

The inclusion of a learnable atom count predictor is a surprisingly impactful addition. Most generative models just sample atom counts from empirical priors, which often leads to pocket-size mismatches. Here, by directly predicting atom counts from protein geometry (volume, surface area, atom count), the model creates molecules that better fit into the pocket. The ablations support this clearly.

The use of conditional flow matching (CFM) for discrete atom types is new and nicely derived. While [8] and others model discrete atoms in diffusion frameworks, this work generalizes the vector field formulation to discrete types in flow matching, which is a non-trivial extension.

On empirical performance, PAFlow leads across most binding affinity metrics (Vina Score, Vina Min, Vina Dock), outperforming strong baselines like ALiDiff and MolCRAFT. It also achieves good QED and SA scores while preserving diversity. The figure showing that PAFlow achieves best performance on 77% of targets is especially compelling.

The sampling efficiency is another win. Despite using ODE-based generation, PAFlow is significantly faster than TargetDiff and Pocket2Mol, and even competes with MolCRAFT when run with fewer steps.

The paper is technically detailed and derivations are thorough. The proofs in the appendix about SE(3) equivariance and the effect of noise injection in the atom number predictor are solid and well-grounded.

While the results are strong, the novelty of the base generation framework is incremental. Flow Matching has already been applied in this space (e.g. FlowSBDD), and the main novelty lies in the protein-guided vector field and atom count handling. That’s still a valuable contribution, but not a foundational shift.

The protein–ligand interaction predictor is trained on normalized binding scores. It would help to discuss how sensitive the model is to this normalization choice, and whether a more realistic affinity predictor (e.g. based on physical interactions) could further improve things.

The molecular property control (QED, SA, etc.) is not explicitly incorporated but still performs reasonably. Since the authors already use a binding affinity predictor to guide generation, adding similar predictors for QED or SA in future work seems like a straightforward and useful extension.

There’s no mention of how well this method handles large or irregular binding pockets. A few edge-case visualizations where PAFlow fails or generates oddly-shaped molecules would help clarify model limitations.

Runtime stats are useful, but it might help to report average time per molecule instead of (or alongside) time for 100 samples; just easier to interpret.

[Final Thoughts]: I have raised my score based on the sufficient author responses.

---

> ### Author Rebuttal · Authors · 2025-07-31
>
> **Q1**:	While the results are strong, the novelty of the base generation framework is incremental. Flow Matching has already been applied in this space (e.g. FlowSBDD), and the main novelty lies in the protein-guided vector field and atom count handling. That’s still a valuable contribution, but not a foundational shift.
>
> **A1**: Thank you for acknowledging our contributions in the design of the prior-guided vector field and atom number prediction. While both PAFlow and FlowSBDD are built upon the Flow Matching (FM) framework, their modeling strategies differ substantially. Flow Matching is a simulation-free approach for stably training continuous normalizing flows (CNFs) that demonstrates strong generative capabilities. Various types of probability paths can be used within the FM framework as summarized in Table 1 of [1], and the choice can be flexibly adapted based on the specific task requirements.
>
> FlowSBDD employs Rectified Flow to learn the transport mapping of atom coordinates and types from the prior to the data distribution, aiming for straight-line trajectories between the two. FlexSBDD adopts a similar modeling approach. While such straight-line paths are computationally efficient, they lack the capacity to model complex generation tasks effectively. In contrast, PAFlow utilizes different probability paths tailored to the distinct characteristics of atom coordinates and types. Specifically, continuous coordinates are modeled using Gaussian distributions with variance-preserving (VP) trajectories derived from diffusion models, while discrete atom types are modeled with categorical distributions, for which we specifically construct a FM formulation and conditional vector field (see lines 144–149 on page 4, detailed derivations in Appendix A.2). In addition, we prove that the resulting generative process is consistent with SE(3)-transformation invariance, offering a theoretical justification for our modeling strategies (see Appendix A.1). We compare these methods under the same experimental settings. As shown in the table, PAFlow significantly outperforms both FlowSBDD and FlexSBDD across all affinity-related metrics while maintaining favorable molecular properties, demonstrating the effectiveness of our modeling approach. Overall, although all these methods fall within the FM framework, they differ fundamentally in the design of probability paths and modeling strategies. We hope these extensions offer useful insights and contribute meaningfully to the advancement of FM-based molecular generation.
>
> | Methods| | Vina Score↓ | Vina Min ↓ | Vina Dock ↓ | High Affinity ↑ | QED ↑ | SA ↑ | Diversity ↑ |
> |-|-|-|-|-|-|-|-|-|
> | TargetDiff      | Avg.        | -5.47                     | -6.64                   | -7.80                    | 58.1%                     | 0.48             | 0.58            | 0.72                   |
> |                 | Med.        | -6.30                     | -6.86                   | -7.91                    | 59.1%                     | 0.48             | 0.58            | 0.71|
> | FlowSBDD| Avg. | -3.62| -6.72| -8.50 | 63.4% | 0.47| 0.51| 0.75|
> |                 | Med.        | -5.03                     | -6.60                   | -8.36                    | 70.9%                     | 0.48             | 0.51            | 0.75                   |
> | FlexSBDD        | Avg.        | -6.64                     | -8.27                   | -9.12                    | 78.5%                     | 0.58             | 0.69            | 0.76                   |
> |                 | Med.        | -7.25                     | -8.46                   | -9.25                    | 84.2%                     | 0.59             | 0.73            | 0.75                   |
> | PAFlow          | Avg.        | **-8.31**                 | **-8.79**               | **-9.46**                | **80.8%**                 | 0.49             | 0.57            | 0.71                   |
> |                 | Med.        | **-8.92**                 | **-8.96**               | **-9.49**                | **93.7%**                 | 0.50             | 0.57            | 0.70                   |
>
>
> **Q2**: The protein–ligand interaction predictor is trained on normalized binding scores. It would help to discuss how sensitive the model is to this normalization choice, and whether a more realistic affinity predictor (e.g. based on physical interactions) could further improve things.
>
> **A2**: Thank you for your valuable suggestion. During the training of the interaction predictor, we apply the widely-used min-max normalization to linearly map the affinity labels into the [0, 1] range. To examine the sensitivity to the choice of normalization, we additionally normalize the labels to the [-1, 1] range by dividing them by the maximum absolute value, and train a new model named PAFlow_abs. The results comparing molecules generated by both methods are shown in the table, where PAFlow_abs achieves comparable performance to PAFlow. For efficiency, PAFlow_abs uses the same scaling factor for guidance as PAFlow. The performance of PAFlow_abs can be further improved by tuning a more suitable scaling factor for its specific normalization. These results suggest that as long as the normalization does not distort the relative distribution of the original labels, the predictor’s effectiveness remains stable. In addition, we agree that using a more realistic predictor based on physical interactions may provide more accurate guidance and lead to improved generation quality. We plan to explore this direction in future work by integrating physical modeling and energy-based prediction.
>
> | Methods| Vina Score ↓| Vina Min ↓| Vina Dock ↓| High Affinity ↑| QED ↑| SA ↑| Diversity ↑|
> |-|-|-|-|-|-|-|-|
> |PAFlow| -8.10| -8.50| -9.62 | 80.7% | 0.49| 0.57| 0.70|
> | PAFlow_abs| -7.74| -8.05| -8.32| 74.6% | 0.43| 0.59| 0.71|
>
> **Q3**: The molecular property control (QED, SA, etc.) is not explicitly incorporated but still performs reasonably. Since the authors already use a binding affinity predictor to guide generation, adding similar predictors for QED or SA in future work seems like a straightforward and useful extension.
>
> **A3**:
> We strongly support your suggestion to extend the guidance for QED and SA.  We conduct preliminary experiments where, in addition to affinity, predictors for QED and SA are trained, and a simple gradient summation is employed to simultaneously guide all three properties. This approach is named PAFlow-multi. The experimental results are shown below, where “backbone” denotes the baseline without any guidance. The Hit Rate represents the percentage of generated molecules that satisfy QED ≥ 0.4, SA ≥ 0.5, and Vina Dock ≤ -8.18 [2]. The results indicate that incorporating the property predictor guidance leads to improvements across all metrics, along with a higher rate of the hit molecules, which demonstrates the effectiveness of the guidance strategy. Furthermore, jointly optimizing binding affinity, QED, and SA is essentially a multi-objective optimization problem, where conflicts among these properties inherently exist. Simply summing the gradients is insufficient to fully address this issue. Recent work (e.g., MoC [3]) has made initial attempts to balance such trade-offs, but this remains an important direction for future research.
>
> | Methods| Vina Score ↓| Vina Min ↓| Vina Dock ↓| High Affinity ↑| QED ↑| SA ↑| Diversity ↑| Hit Rate ↑|
> |-|-|-|-|-|-|-|-|-|
> |backbone|-5.24|-6.90|-8.30|64.7%|0.52|0.56|0.71|24.9%|
> | PAFlow-multi | **-6.73** | **-7.74**|**-8.75**|**75.1%**|**0.53**|**0.58**| **0.72**|**29.3%**|
>
>
> **Q4**: There’s no mention of how well this method handles large or irregular binding pockets. A few edge-case visualizations where PAFlow fails or generates oddly-shaped molecules would help clarify model limitations.
>
> **A4**: Thank you for your valuable suggestion. We fully agree with your point. Since the OpenReview does not allow uploading images, we will supplement the visualizations of failure cases or irregularly structured molecules generated by PAFlow to provide a clearer illustration of its limitations in the revision.
>
> **Q5**: Runtime stats are useful, but it might help to report average time per molecule instead of (or alongside) time for 100 samples; just easier to interpret.
>
> **A5**: Thank you for your suggestion. We report the average generation time per molecule as a supplement to Fig. 4. We hope this table provides a more intuitive comparison across different methods and further highlights the high efficiency of PAFlow.
>
> |Methods|Pocket2Mol|TargetDiff|MolCRAFT|PAFlow|PAFlow T=20|
> |-|-|-|-|-|-|
> | **Avg Time (s)** |39.7|40.1|4.0|7.2|**2.9**|
>
> [1] Tong A, Fatras K, Malkin N, et al. Improving and generalizing flow-based generative models with minibatch optimal transport[J]. arXiv preprint arXiv:2302.00482, 2023.
>
> [2] Dorna V, Subhalingam D, Kolluru K, et al. Tagmol: Target-aware gradient-guided molecule generation[J]. arXiv preprint arXiv:2406.01650, 2024.
>
> [3] Zhou J, Zhao D, Qian H, et al. Multi-Objective Structure-Based Drug Design Using Causal Discovery[J]. IEEE Transactions on Computational Biology and Bioinformatics, 2025.

---

> > ### Comment · Reviewer_AuA9 · 2025-08-05
> >
> > Thank you for the detailed and well-organized rebuttal. The additional context around the modeling choices in PAFlow, particularly the design of distinct probability paths for continuous and discrete variables, helps clarify how your method meaningfully extends beyond prior FM-based approaches like FlowSBDD. I appreciate the quantitative comparisons as well as the theoretical justifications around SE(3) invariance.
> >
> > Your experiments on normalization sensitivity for the interaction predictor and preliminary results on QED/SA-guided generation are thoughtful and reinforce the generality of your framework. The discussion on future directions, especially multi-objective optimization beyond gradient summation is well taken.
> >
> > Overall, the response addresses all the concerns I raised. I maintain my original score and look forward to the revised manuscript.

---

> > > ### Author Response · Authors · 2025-08-06
> > >
> > > Thank you sincerely for your reply and recognition of our efforts. We appreciate your insightful suggestions and will include the corresponding discussions and results in the final version. Please feel free to contact us if you have any other questions.

---

### Note · Authors · 2025-08-12

We sincerely thank all reviewers and the AC for their time and constructive feedback! We have carefully addressed all concerns raised by the reviewers, providing detailed clarifications supported by both theoretical analysis and experimental evidence in our rebuttal and discussion. The key issues identified—including the distinctions between our method and FlexSBDD, PocketFlow, and TAGMol, the evaluation of structural metrics for generated molecules, and the effectiveness of the atom number predictor—have been comprehensively clarified.

In this work, we propose PAFlow, a flow matching-based framework that models continuous atom coordinates using Variance Preserving Conditional Flow Matching (CFM) and discrete atom types using a newly designed CFM, with theoretical guarantees for SE(3)-invariance. We further employ a protein–ligand interaction predictor to guide the generative process toward higher affinity, and introduce an atom number predictor to generate ligands whose sizes better align with the geometry of the protein pocket. Experimental results demonstrate that PAFlow achieves state-of-the-art binding affinity while maintaining favorable molecular properties and stable structures. Our contributions have been recognized by reviewers during the discussion phase.

We believe that the raised concerns have been fully addressed, and that our work offers both practical significance and potential impact for structure-based drug design. We hope that these clarifications and contributions will support a positive final assessment of our work.

---

### Decision · Program_Chairs · 2025-09-17

**Decision:**

Accept (poster)

**Comment:**

The paper introduces a target-aware generative model using flow matching for structure-based drug design (SBDD). The proposed approach has shown to be effective through relatively solid empirical evaluations. There are shared concerns regarding relatively incremental methodology contribution as flow-matching has already been used in solving SBDD, such as FlowSBDD, FlexSBDD and PocketFlow. The author provided additional results and resolved most of the concerns. While reviewer **ns3U** remain less enthusiastic regarding the limited novelty, the rebuttal convinced two reviewers to raise their scores. The AC tends to be on the positive side and believes the paper contributes valuable insights and effective approach to the ML and SBDD community, thus recommending acceptance. The AC has discussed the case with the SAC, and that the decision has been confirmed by the SAC. As suggested by the reviewers, the final version should include more detailed clarifications and discussion to better highlight the model innovation.